# Empirical Gaussian Processes

**Jihao Andreas Lin** [* 1] **Sebastian Ament** [* 1] **Louis C. Tiao** [1] **David Eriksson** [1] **Maximilian Balandat** [1] **Eytan Bakshy** [1]

## Abstract

Gaussian processes (GPs) are powerful and widely used probabilistic regression models, but their effectiveness in practice is often limited by the choice of kernel function. This kernel function is typically handcrafted from a small set of standard functions, a process that requires expert knowledge, results in limited adaptivity to data, and imposes strong assumptions on the hypothesis space. Re-evaluating this challenge from a hierarchical Bayesian and function-space view, we study Empirical GPs, a principled framework for constructing flexible, data-driven GP priors that overcome these limitations. Rather than relying on standard parametric kernels, we estimate the mean and covariance functions empirically from a corpus of historical observations, enabling the prior to reflect rich, non-trivial covariance structures present in the data. Theoretically, we show that the resulting model converges to the GP that is closest (in KL-divergence sense) to the real data-generating process. We formulate the problem of learning the GP prior from independent datasets as maximum likelihood estimation and derive an Expectation-Maximization algorithm with closed-form updates, allowing the model handle heterogeneous observation locations across datasets. We demonstrate that Empirical GPs achieve competitive performance on learning curve extrapolation and time series forecasting benchmarks.

## 1. Introduction

Gaussian processes (GPs) provide an elegant framework for probabilistic regression, offering data efficiency and principled uncertainty quantification essential for applications ranging from scientific modeling to Bayesian optimization

---
[*]Equal contribution [1]Meta. Correspondence to: Jihao Andreas Lin <jandylin@meta.com>, Sebastian Ament <sebastianament@meta.com>.

*Proceedings of the $43^{rd}$ International Conference on Machine Learning*, Seoul, South Korea. PMLR 306, 2026. Copyright 2026 by the author(s).

(Rasmussen & Williams, 2006). However, the practical utility of GPs hinges on the choice of kernel function, which encodes prior beliefs about the function being modeled. The kernel function is typically selected from a small set of standard stationary kernels such as radial basis functions (RBF), Matérn, or periodic kernels, and the kernel hyperparameters are estimated by maximizing the marginal log-likelihood.

This approach is limited in its flexibility, since no single kernel is appropriate across all settings. Having an expert hand-craft an artisan kernel can substantially improve the model, but is only feasible for the most critical applications. Duvenaud et al. (2013) and others have tried to overcome this by automatically constructing kernel compositions in a data-driven fashion, but this is still limited to sums and products of canonical kernels, and requires approximations to intractable integrals. Finally, many standard kernels are stationary, which are poorly suited for extrapolation, i.e. predictions at inputs beyond the training range. The extrapolation problem is common to numerous practical settings, such as predicting the final performance of a machine learning model after observing only the first few epochs of training (Swersky et al., 2014; Lin et al., 2025). Learning curves typically exhibit a characteristic structure of rapid initial improvement followed by gradual saturation, which is consistent across experiments but not captured by stationary kernels. Similar challenges arise in time series forecasting, where future predictions must extrapolate temporal patterns, and in sequential experimental design, where expensive evaluations must be guided by predictions at unexplored configurations outside the support of the experimental data.

Recent work used historical data to construct more informative GP priors. Perrone et al. (2018); Wistuba & Grabocka (2021) consider using neural networks as feature extractors or deep kernels for knowledge transfer in Bayesian optimization. Wang et al. (2024) proposed pre-training neural networks to parameterize mean and covariance functions for GPs using historical HPO (hyperparameter optimization) datasets. These approaches require careful architecture design, large training corpora to avoid overfitting, and can require time-consuming hyperparameter optimization.

In this paper, we study *Empirical Gaussian Processes*, a framework that takes a fundamentally different approach. Rather than learning hyperparameters of a parametric ker-

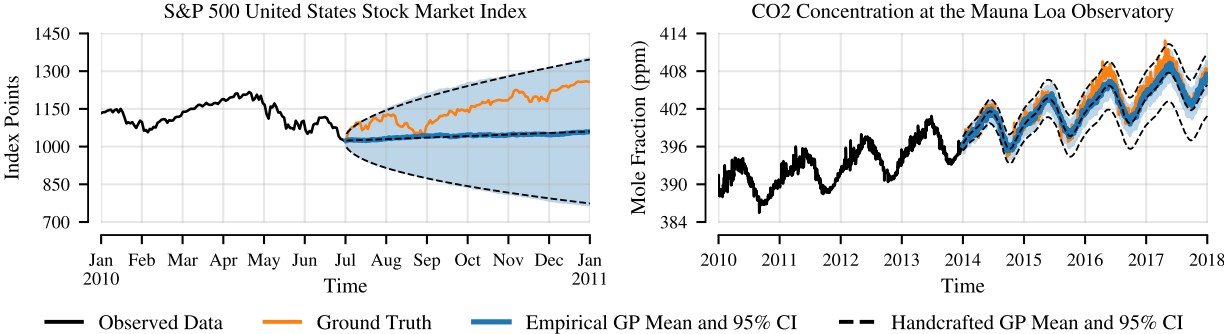

*Figure 1.* Without human intervention, Empirical GP captures the behavior of kernels handcrafted by human experts. **Left:** On financial stock market data, Empirical GP matches the canonical geometric Brownian motion model. While it cannot be expected to predict future prices exactly, it recovers the optimal statistical variance and drift of the data-generating process. **Right:** On atmospheric climate data, Empirical GP infers seasonality and an upwards trend, achieving 21% lower RMSE than an expert-designed kernel (Rasmussen, 2024).

nel, we estimate the GP prior mean and covariance functions directly from a corpus of historical observations of the data-generating stochastic process. The premise of this approach is that available independent realizations, e.g., learning curves from previous experiments, serve as samples from the unknown prior. The empirical prior, properly estimated, will therefore capture structure present in the historical data, including non-stationarity, heteroscedasticity, and domain-specific correlations. Unlike parametric kernels that impose rigid assumptions, the empirical covariance adapts to the data, enabling extrapolation that adheres to patterns observed in previous realizations of the process.

Our work revisits the Hierarchical Bayesian framework of Schwaighofer et al. (2004), which we extend by generalizing the training algorithm to support spatially heterogeneous observations without grid alignment, and by introducing a novel inference scheme that prevents overconfidence when extrapolating away from historical data.

Our main contributions are as follows:

1. We introduce Empirical GPs, a principled Bayesian model for learning non-parametric GP priors from independent datasets via maximum likelihood estimation.

2. We provide function-space-theoretical justification, proving that Empirical GPs converge to the best Gaussian approximation of the true data-generating process.

3. We derive a closed-form Expectation-Maximization algorithm with exact E- and M-steps, extending prior work from the discrete, fixed-grid setting to continuous domains with sparse, irregularly distributed observations.

4. We develop a technique to extend the learned prior to new inputs, analogous to inducing point methods but operating directly on the prior to avoid variance starvation.

5. We demonstrate competitive predictive performance on time series forecasting, learning curve extrapolation, and

ML hyper-parameter modeling problems, outperforming multiple optimization-heavy baselines.

## 2. Related Work

**Meta-Learning for Gaussian Processes.** Several approaches learn GP priors from multi-task data. Zhou & Precioso (2019) learn basis functions for Bayesian linear regression, while Rothfuss et al. (2021) provide PAC-Bayesian guarantees for meta-learning GP hyperparameters. These methods optimize within parametric kernel families. In contrast, we follow the idea of Schwaighofer et al. (2004) and directly estimate the prior mean and covariance functions from sample statistics, yielding a non-parametric prior.

**Multi-Task GPs.** Multi-task GP methods model correlations between tasks through task kernels (Bonilla et al., 2007; Swersky et al., 2013), but scale poorly with the number of tasks. Transfer learning approaches (Salinas et al., 2020b; Feurer et al., 2018) offer improved scalability but sacrifice principled uncertainty quantification.

**Variational Inference.** Our formulation shares similarities with variational GP methods (Titsias, 2009; Hensman et al., 2013): both work with finite-dimensional marginals. However, variational methods approximate a posterior given a fixed prior, whereas we estimate the prior itself.

**Expressive Kernels.** Deep kernel learning (Wilson et al., 2016) learns flexible features but requires substantial amounts of data and model fitting can suffer from instabilities. Spectral mixture kernels (Wilson & Adams, 2013) can extrapolate quasi-periodic patterns but assume stationarity. Our empirical approach sidesteps these choices: the covariance is estimated directly from data, inheriting whatever structure exists in historical observations.

**Learning Curve Prediction.** Domhan et al. (2015) extrapolate learning curves using parametric models, while Swersky

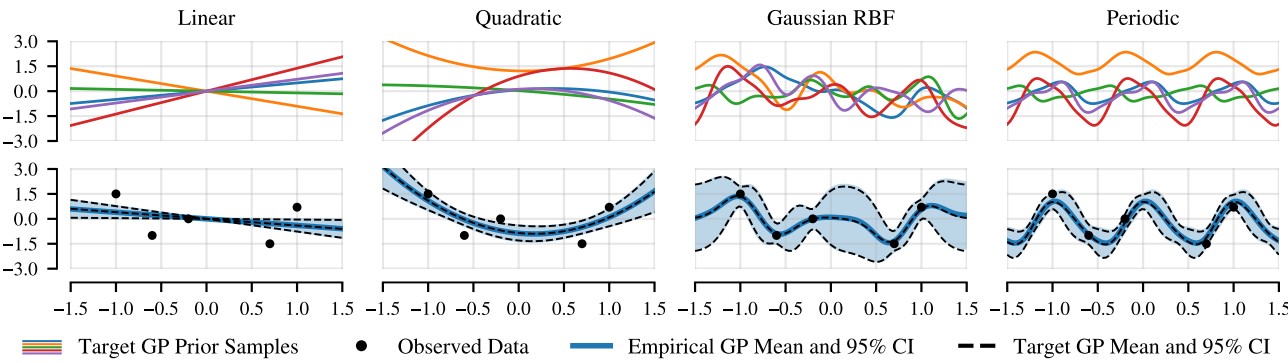

*Figure 2.* **Top:** Sample paths from a variety of target GPs with distinct covariances functions are used as historical data for Empirical GP. **Bottom:** On a fixed set of new observations, the Empirical GP posterior converges to the posterior of the corresponding target GP, when using 256 sample paths from the latter as historical data, demonstrating data-driven adaptivity and convergence to the target GP.

et al. (2014) use a GP with a custom exponential mixture kernel to extrapolate learning curves. Lin et al. (2025) leverage latent Kronecker structure to scalably and jointly model hyper-parameter and learning curve progression dimensions. Adriaensen et al. (2023) train Prior-Fitted Networks (PFNs) on learning curve data sampled from parametric priors.

# 3. Empirical Gaussian Processes

## 3.1. Preliminaries

A Gaussian Process (GP) is a distribution over functions $g$, any finite marginal of which has a joint multivariate normal distribution. A GP is fully specified by its mean function $m(\mathbf{x}) = \mathbb{E}[g(\mathbf{x})]$ and its covariance function (kernel) $k(\mathbf{x}, \mathbf{x}') = \mathbb{E}[(g(\mathbf{x}) - m(\mathbf{x}))(g(\mathbf{x}') - m(\mathbf{x}'))]$. We write:

$$g(\mathbf{x}) \sim \mathcal{GP}\left(m(\mathbf{x}), k(\mathbf{x}, \mathbf{x}')\right).$$

Any valid kernel function must be positive semi-definite (PSD). That is, for any set of input points $\{\mathbf{x}_1, \ldots, \mathbf{x}_N\} \subset \mathcal{X}$ and any non-zero vector $\mathbf{v} \in \mathbb{R}^N$, the kernel matrix $\mathbf{K}$ with entries $K_{ij} = k(\mathbf{x}_i, \mathbf{x}_j)$ must satisfy $\mathbf{v}^\top \mathbf{K} \mathbf{v} \geq 0$.

## 3.2. Function-Space View

Suppose we want to approximate a data-generating stochastic process. Given $S$ independent sample paths $f_1, \ldots, f_S$, we can estimate their true mean $m = \mathbb{E}[f]$ and covariance function $k = \text{Cov}(f)$ via maximum likelihood,

$$m_S(\mathbf{x}) = \frac{1}{S} \sum_{i=1}^{S} f_i(\mathbf{x}), \quad k_S(\mathbf{x}, \mathbf{x}') = \frac{1}{S} \sum_{i=1}^{S} \tilde{f}_i(\mathbf{x}) \tilde{f}_i(\mathbf{x}'), \tag{1}$$

where $\tilde{f}_i(\mathbf{x}) = f_i(\mathbf{x}) - m(\mathbf{x})$. We define the *Empirical* GP using the *empirical* mean $m_S$ and covariance function $k_S$ as $\mathcal{GP}\left(m_S, k_S\right)$. Note that $k_S$ is a valid kernel by construction, as it is a sum of outer products of the centered functions.

## 3.3. Theoretical Results

We now provide a selection of key theoretical results that ground our approach. A more formal treatment including all assumptions and proofs is given in Appendix A.

Let $\mathcal{GP}\left(m_S, k_S\right)$ be the Empirical GP defined by the empirical mean function $m_S$ and empirical covariance function $k_S$, conditioned on observed sample paths $f_1, ..., f_S$. Additionally, let $\mathcal{GP}\left(m, k\right)$ be a GP defined by the true mean function $m$ and true covariance function $k$ of $f$.

**Proposition 1.** *Assume that $k$ is continuous and that its canonical semi-metric satisfies Dudley's entropy integral condition. Then, for almost every sequence of sample paths $\{f_i\}_{i=1}^{S}$, we have $\mathcal{GP}\left(m_S, k_S\right) \rightharpoonup \mathcal{GP}\left(m, k\right)$ as $S \to \infty$.*

In particular, the limit $\mathcal{GP}\left(m, k\right)$ is the best possible Gaussian approximation of the true underlying stochastic process in terms of the KL divergence definition by Sun et al. (2019).

**Proposition 2.** *Let $\mathbb{P}$ denote the law of $f$, and let $\mathbb{G}$ be any GP indexed by $\mathcal{X}$. Assume that $\mathbb{P}$ is absolutely continuous with respect to $\mathbb{G}$, and that $k$ is strictly positive definite. Then, $\lim_{S \to \infty} \mathcal{GP}\left(m_S, k_S\right) = \arg\min_{\mathbb{G}} D_{\text{KL}}(\mathbb{P} \parallel \mathbb{G})$.*

Proposition 2 provides firm theoretical grounding for the use of Empirical GPs as a general modeling approach for learning from historical data.

## 3.4. Discrete Observations

In practice, we cannot observe full continuous sample paths. Real-world data arrives as finite, discrete sets of observations $\{\mathbf{X}_i, \mathbf{y}_i\}$, where the input locations $\mathbf{X}_i$ may vary arbitrarily from sample to sample.

Important special cases are *densely sampled and low-dimensional settings*—such as high-frequency time series—where we can approximate the continuous statistics via interpolation. Formally, for each sample $i$, we define

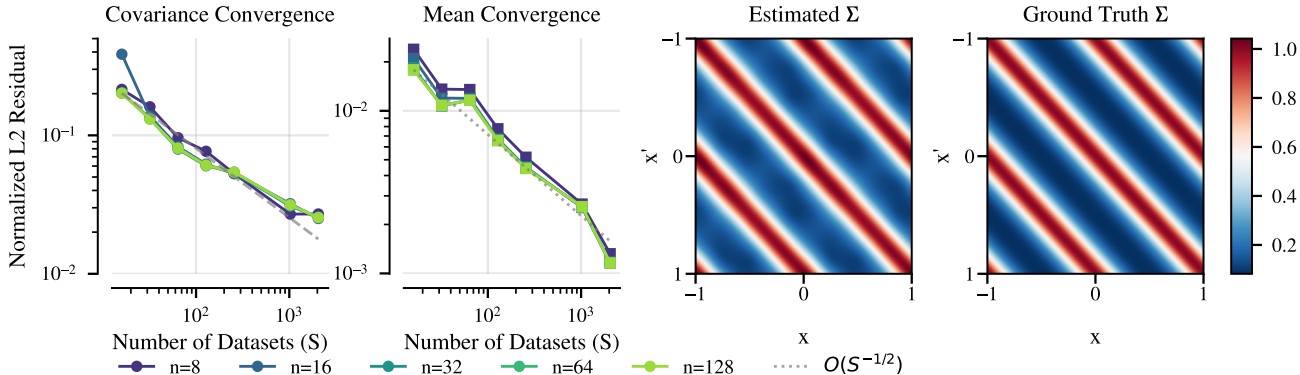

*Figure 3.* **Left:** Convergence of the EM-estimated prior mean and covariance to the ground truth, as a function of the number of independent incomplete datasets (S), and for different numbers of observed data points per dataset (n). **Right:** The estimated covariance matrix ($S = 1024, n = 64$) follows the ground truth closely.

an (e.g. linear) interpolant $\tilde{f}_i(\cdot) = \mathcal{I}(\cdot; \mathbf{X}_i, \mathbf{y}_i)$, and treat $\tilde{f}_i$ as fully observed realizations of the process, allowing us to evaluate the empirical covariance in (1) directly at *arbitrary* input locations. Despite its simplicity, this heuristic leads to surprisingly strong results; as shown in Sec. 4.3, this interpolation-based Empirical GP outperforms several deep learning models on a time series forecasting benchmark.

**Efficient Computation via SVD**  While evaluating (1) is efficient, the cost scales linearly with the number of historical samples $S$. To further accelerate inference for large datasets, we discretize the interpolants onto a shared reference grid $\mathbf{Z} = \{\mathbf{z}_1, \cdots, \mathbf{z}_M\}$ to form a matrix $\mathbf{Y} \in \mathbb{R}^{S \times M}$ of centered observations. We then employ a lossless compression using an SVD, $\mathbf{Y} = \mathbf{U}\mathbf{S}\mathbf{V}^\top$, constructing a reduced set of $M$ "eigen-observations" $\tilde{\mathbf{Y}} = \mathbf{S}\mathbf{V}^\top \in \mathbb{R}^{M \times M}$. Interpolating the scaled eigen-observations $\tilde{v}_j(\cdot) = \mathcal{I}(\cdot; \mathbf{Z}, [\tilde{\mathbf{Y}}]_j)$ enables kernel evaluation at arbitrary locations while reducing complexity from $\mathcal{O}(S)$ to $\mathcal{O}(M)$ by replacing the full dataset with $M$ basis functions.

### 3.5. Continuous-Domain Empirical Gaussian Process

We now address the more general and challenging scenario where observations are sparse and irregularly distributed. In these regimes, geometric interpolation is ill-defined, and sample statistics cannot be computed directly. Instead, we employ a kernel-based mechanism to probabilistically project sparse data onto shared latent variables. This establishes a rigorous generative model that couples the disparate observation sets $\mathbf{X}_i$ through a unified latent structure.

To formalize this framework, we anchor the latent process on a fixed set of $M$ reference locations $\mathbf{Z} = \{\mathbf{z}_1, \ldots, \mathbf{z}_M\} \subset \mathcal{X}$. We treat the corresponding function values $\mathbf{u} = f(\mathbf{Z}) \in \mathbb{R}^M$ as the latent variables. For each historical sample $i$, we observe a vector $\mathbf{y}_i$ at a unique set of input locations $\mathbf{X}_i$ of size $N_i$. Our goal is to recover the population parame-

ters $\boldsymbol{\mu}$ and $\boldsymbol{\Sigma}$ of the latent variables $\mathbf{u}$ that best explain the observations $\{\mathbf{X}_i, \mathbf{y}_i\}_{i=1}^S$.

**Interpolation Mechanism**  To relate the latent variables $\mathbf{u}$ at $\mathbf{Z}$ to observations at arbitrary locations $\mathbf{X}_i$, we use kernel interpolation based on a stationary base kernel $k_{\text{base}}(\cdot, \cdot)$. We define the weight matrix $\mathbf{W}_i \in \mathbb{R}^{N_i \times M}$ as $\mathbf{W}_i = k_{\text{base}}(\mathbf{X}_i, \mathbf{Z}) \, k_{\text{base}}(\mathbf{Z}, \mathbf{Z})^{-1}$.

**Probabilistic Model**  We assume the latent reference values $\mathbf{u}_i$ for each task are drawn from a shared multivariate normal prior:

$$\mathbf{u}_i \sim \mathcal{N}(\boldsymbol{\mu}, \boldsymbol{\Sigma}), \quad i = 1, \ldots, S.$$

The observations $\mathbf{y}_i$ are modeled as the projected latent values with additive Gaussian noise:

$$\mathbf{y}_i \mid \mathbf{u}_i \sim \mathcal{N}(\mathbf{W}_i \mathbf{u}_i, \sigma^2 \mathbf{I}),$$

where $\sigma^2$ is the variance of the measurement noise. This formulation decouples the shared latent structure on $\mathbf{Z}$ from the task-specific observation locations $\mathbf{X}_i$, allowing us to aggregate information from spatially heterogeneous datasets.

**E-step (Imputation)**  We compute the posterior distribution of the latent variables $\mathbf{u}_i$ given the partial observations $\mathbf{y}_i$ and current parameter estimates $\boldsymbol{\mu}^{(t)}, \boldsymbol{\Sigma}^{(t)}$. This posterior is Gaussian:

$$p(\mathbf{u}_i \mid \mathbf{y}_i) = \mathcal{N}(\mathbf{m}_i, \mathbf{C}_i).$$

To streamline the notation, we define the *cross-covariance* between the reference grid and the observation locations as $\boldsymbol{\Sigma}_{\mathbf{Z}, \mathbf{X}_i} = \boldsymbol{\Sigma}^{(t)} \mathbf{W}_i^\top$. The moments are computed as:

$$\mathbf{m}_i = \boldsymbol{\mu}^{(t)} + \boldsymbol{\Sigma}_{\mathbf{Z}, \mathbf{X}_i} \boldsymbol{\Sigma}_{\mathbf{X}_i, \mathbf{X}_i}^{-1} \left( \mathbf{y}_i - \mathbf{W}_i \boldsymbol{\mu}^{(t)} \right),$$

$$\mathbf{C}_i = \boldsymbol{\Sigma}^{(t)} - \boldsymbol{\Sigma}_{\mathbf{Z}, \mathbf{X}_i} \boldsymbol{\Sigma}_{\mathbf{X}_i, \mathbf{X}_i}^{-1} \boldsymbol{\Sigma}_{\mathbf{X}_i, \mathbf{Z}},$$

where $\Sigma_{\mathbf{X}_i, \mathbf{X}_i} = \mathbf{W}_i \Sigma \mathbf{W}_i^\top + \sigma^2 \mathbf{I}$ is the predicted covariance at the observed locations. This highlights that the update is a standard correction term weighted by the correlation between the latent grid and the specific observations.

**M-step (Maximum Likelihood)** We update the parameters by maximizing the expected log-likelihood. The updates simply aggregate the sufficient statistics from all tasks:

$$\boldsymbol{\mu}^{(t+1)} = \frac{1}{S} \sum_{i=1}^{S} \mathbf{m}_i, \qquad \Sigma^{(t+1)} = \frac{1}{S} \sum_{i=1}^{S} \mathbf{C}_i + \tilde{\mathbf{m}}_i \tilde{\mathbf{m}}_i^\top,$$

where $\tilde{\mathbf{m}}_i = \mathbf{m}_i - \boldsymbol{\mu}^{(t+1)}$.

**Computational Complexity** The computational cost of the EM iterations is determined by the size of the reference set $M$ and the per-task observation count $N_i$. Computing the interpolation weights $\mathbf{W}_i$ requires $\mathcal{O}(N_i M^2)$. In the E-step, forming the predictive covariance $\Sigma_{\mathbf{X}_i, \mathbf{X}_i}$ and inverting it requires $\mathcal{O}(N_i M^2 + N_i^3)$ operations. The M-step is dominated by the summation of the conditional covariances $\mathbf{C}_i$, which is $\mathcal{O}(SM^2)$. Thus, the algorithm's complexity is dominated by the E-step $\mathcal{O}(S(N_i M^2 + N_i^3))$. For datasets with large $N_i > M$, one can exploit the Woodbury identity to reduce the inversion cost to $\mathcal{O}(M^3 + N_i M^2)$.

**Residual Interpolation and Base Kernel** The kernel-interpolation described above restricts the learned correlations to a neighborhood of the reference set $\mathbf{Z}$. Consequently, at test locations $\mathbf{X}_*$ far from $\mathbf{Z}$ where the interpolation weights vanish ($\mathbf{W}_* \to \mathbf{0}$), the predictive variance collapses to the noise floor. This leads to exceedingly high confidence in regions where the model should exhibit high epistemic uncertainty, that is, variance starvation.

To resolve this, we employ a residual interpolation scheme. Instead of assuming the latent variables $\mathbf{u}$ capture the total function, we interpret the learned parameters $\boldsymbol{\mu}$ and $\Sigma$ on the grid $\mathbf{Z}$ as non-parametric adjustments ("shifts") to a base parametric mean function $\mu_{\text{base}}(\cdot)$ and kernel $k_{\text{base}}(\cdot, \cdot)$. We define the learned residuals at the reference locations as:

$$\boldsymbol{\delta}_\mu = \boldsymbol{\mu} - \mu_{\text{base}}(\mathbf{Z}), \qquad \boldsymbol{\delta}_\Sigma = \Sigma - k_{\text{base}}(\mathbf{Z}, \mathbf{Z}).$$

These quantities capture the structural, non-stationary deviations that the base model fails to explain. To generalize to new inputs $\mathbf{x}$, we project these residuals through the base kernel structure:

$$\begin{aligned} \mu(\mathbf{x}) &= \mu_{\text{base}}(\mathbf{x}) + \mathbf{W}_{\mathbf{x}} \boldsymbol{\delta}_\mu, \\ k(\mathbf{x}, \mathbf{x}') &= k_{\text{base}}(\mathbf{x}, \mathbf{x}') + \mathbf{W}_{\mathbf{x}} \boldsymbol{\delta}_\Sigma \mathbf{W}_{\mathbf{x}'}^\top, \end{aligned} \qquad (2)$$

where $\mathbf{W}_{\mathbf{x}} = k_{\text{base}}(\mathbf{x}, \mathbf{Z}) k_{\text{base}}(\mathbf{Z}, \mathbf{Z})^{-1}$.

Therefore, the base kernel $k_{\text{base}}$ does not serve as a rigid assumption or a noise floor. Instead, this formulation explicitly decouples two complementary types of behaviors:

1) *Within-Domain Extrapolation:* Within the range of the historical support, Empirical GPs use the learned non-stationary covariances to extrapolate sample paths without reverting prematurely to an uninformative prior mean.

2) *Off-Grid Robustness:* Far from any historical coordinates, where $\mathbf{W}_{\mathbf{x}} \to \mathbf{0}$, the prior gracefully reverts to the baseline behavior of $k_{\text{base}}$. While this introduces a dependency on $k_{\text{base}}$, the base mean and kernel can simple be chosen and optimized as a canonical GP, and then used for the EM-algorithm above. This straightforward approach is sufficient to outperform competing methods on a 7-dimensional hyper-parameter modeling problem in Section 4.5.

**Residual Interpolation in Training** Leveraging residual interpretation also requires a modification of the E-step. Treating the projection $\mathbf{W}_i \mathbf{u}_i$ as exact ignores the base-kernel variance at $\mathbf{X}_i$ not captured by $\mathbf{Z}$. Instead, we set

$$\mathbf{Q}_i = \left( \mathbf{K}_{\mathbf{X}_i \mathbf{X}_i} - \mathbf{W}_i \mathbf{K}_{\mathbf{Z}\mathbf{Z}} \mathbf{W}_i^\top \right) + \sigma^2 \mathbf{I},$$

so that $\mathbf{y}_i \mid \mathbf{u}_i \sim \mathcal{N}(\mathbf{W}_i \mathbf{u}_i, \mathbf{Q}_i)$.

**Relation to Inducing Points** Equation (2) has similarities to variational inducing-point methods (Titsias, 2009), but with a crucial difference: we use it to *define the prior*, whereas variational methods use an analogous expression to *approximate the posterior* under a fixed prior. Consequently, the variational posterior mean is confined to a rank-$M$ subspace, while the posterior mean induced by the Empirical GP prior can occupy an arbitrarily high-rank space with an appropriate base kernel.

### 3.6. Related Approaches

Schwaighofer et al. (2004) similarly employ an EM for non-parametric covariance learning, compared to which our approach introduces two critical advancements. First, Schwaighofer et al. (2004) rely on a fixed-grid discretization, while the probabilistic interpolation mechanism above enables training directly on spatially heterogeneous, continuous observations without alignment. Second, we address the *variance starvation* inherent in their Nyström-based prediction, where uncertainty collapses to zero away from the reference set, ensuring robust uncertainty quantification that reverts gracefully to the base prior in unexplored regions.

HyperBO (Wang et al., 2024) optimizes a Deep Kernel GP ($k_\theta(\mathbf{x}, \mathbf{x}') = k_\nu(\phi_\theta(\mathbf{x}), \phi_\theta(\mathbf{x}'))$) via gradient descent. While expressive, the learned correlations are implicit within the network weights $\theta$, making the prior difficult to interpret. In contrast, our EM approach explicitly estimates the covariance matrix $\Sigma$ on reference inputs, and enables stable, closed-form updates, avoiding the extensive iterations and instabilities of gradient-based methods.

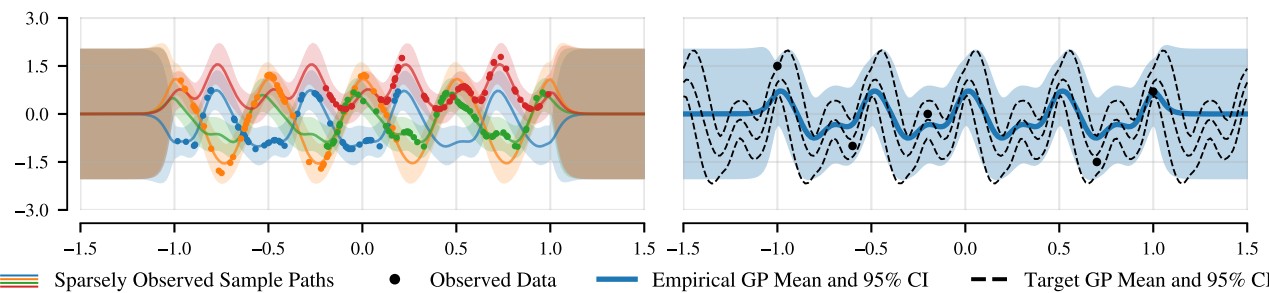

*Figure 4.* **Left:** The EM-inferred conditional distributions for each set of independent historical observations (points). The historical inputs are sampled uniformly from one of two continuous intervals: (-1, 0.2) and (-0.2, 1). The EM-inferred conditional distributions exhibit the periodic characteristics beyond the respective interval associated with a given set of historical observations, and revert gracefully to the base kernel (Matérn) beyond the range of any historical observations (-1, 1). **Right:** The posterior distribution associated with the EM-inferred GP prior (blue), compared to the ground truth posterior (dashed). The Empirical GP closely follows the ground truth within the interval of historical observations (-1, 1), and reverts back to the base kernel's behavior beyond.

## 4. Experiments

We first qualitatively demonstrate that Empirical GPs recover the behavior of handcrafted kernels directly from data. Next, we verify our theoretical results, showing that Empirical GPs converge to the target stochastic process when it is Gaussian (see Proposition 1 and Corollary 1). We then present a quantitative evaluation on the GIFT-Eval time series forecasting benchmark (Aksu et al., 2024), followed by learning curve extrapolation using data from LCBench (Zimmer et al., 2021). Our Empirical GP model is implemented in BoTorch (Balandat et al., 2020).

### 4.1. Capturing the Behavior of Handcrafted Kernels

Our first task considers S&P 500 stock market data. The canonical human-expert model for this type of data is geometric Brownian motion, which assumes that the logarithm of the stock price follows a random walk with drift, $dS_t = \mu S_t dt + \sigma S_t dW_t$, where $S_t$ is the stock price at time $t$, $\mu$ is the expected rate of return, $\sigma$ is the volatility, and $W_t$ is Brownian motion. For our experiment, we calculate the parameters $\mu$ and $\sigma$ on the same historical data which is used by our Empirical GP.

The second task considers the Mauna Loa carbon dioxide concentration dataset (Thoning et al., 2025). We implement a custom kernel which was handcrafted for this dataset by a human expert (Rasmussen, 2024). It consists of three additive components which respectively model the trend, seasonality, and noise. The trend component is a sum of once, twice and thrice integrated white noise, the seasonal contribution is modeled by a product of a periodic kernel with a Gaussian RBF kernel, and the noise kernel consists of a Gaussian RBF kernel with additive homoskedastic noise.

Since both tasks consist of extrapolating a single time series, we use sliding windows to extract additional subseries which can be used as historical data for Empirical GP. For

the financial data, we extracted subseries between Jan 1st, 1930 and Jan 1st 2010, and for the climate data, we considered the time span from Jan 1st, 1975 to Jan 1st, 2010. Intuitively, this procedure assumes self-similarity in the underlying process, which, for example, is indeed the case for geometric Brownian motion. Additionally, for the financial data, we apply a log transform and align subseries to start at zero, to reflect its exponential nature. In other settings where multiple data sets from the same process are available (e.g. learning curves, climate data from similar geographic locations) this procedure is unnecessary.

Figure 1 illustrates that the Empirical GP recovers the behavior of the handcrafted kernels. On the stock market data, the Empirical GP almost perfectly matches geometric Brownian motion, despite not being (explicitly) aware of the model and its ground truth parameters. In light of Proposition 2, this can be interpreted as validating geometric Brownian motion as the best possible Gaussian approximation for stock market data. On the atmospheric climate data, the Empirical GP implicitly captures seasonality and the upwards trend, without any explicit inductive bias. Furthermore, compared to the handcrafted kernel, the Empirical GP achieves 21.52% lower RMSE and has 14.11% higher likelihood of generating the ground truth data in the prediction window.

### 4.2. Convergence to Target Gaussian Processes

Proposition 1 suggests that, as the number of historical sample paths increases, the Empirical GP converges to the underlying stochastic process if that process is Gaussian. Furthermore, Corollary 1 states that the corresponding posteriors on new observations also match. We verify these claims empirically by drawing synthetic sample paths from various target GPs with different ground truth covariance functions, namely linear, quadratic, Gaussian RBF, and periodic. Using these synthetic sample paths as historical data for the Empirical GP, we compare its posterior to the posteriors of the target GPs on a fixed set of new observations.

Figure 2 shows that the Empirical GP makes distinct predictions on the same fixed set of new observations when given different sample paths from different target GPs as historical data. In particular, the Empirical GP posterior closely matches the posterior of the corresponding target GP. This demonstrates both the data-driven adaptivity and convergence properties of the Empirical GP.

### 4.3. GIFT-Eval Time Series Forecasting Benchmark

For a quantitative evaluation, we consider the GIFT-Eval time series forecasting benchmark (Aksu et al., 2024), which consists of 97 datasets, spanning seven domains, ten frequencies, and a total of 144,000 time series with 177 million data points. The seven domains are Econ/Fin, Energy, Healthcare, Nature, Sales, Transport, and Web/CloudOps, and each dataset has a fixed prediction length which is associated with short, medium, or long-term forecasts. We do not consider the additionally provided pre-training dataset.

We consider the same baseline methods as Aksu et al. (2024), incorporating the statistical models Naive, Seasonal Naive (Hyndman & Athanasopoulos, 2018), Auto Arima, Auto ETS, and Auto Theta (Garza et al., 2022). We also include a GP with Gaussian RBF kernel and a GP using the spectral mixture kernel with four components (Wilson & Adams, 2013). Furthermore, following Aksu et al. (2024), we also include eight deep learning baselines, namely DeepAR (Salinas et al., 2020a), TFT (Lim et al., 2021), TiDE (Das et al., 2023), N-BEATS (Oreshkin et al., 2020), PatchTST (Nie et al., 2023), DLinear (Zeng et al., 2023), Crossformer (Zhang & Yan, 2023), and iTransformer (Liu et al., 2024).

Due to the heterogeneity of datasets in the GIFT-Eval benchmark, we extract subseries to be used as historical data using a sliding window approach akin to Section 4.1. In particular, we set the window size to be the sum of context length and prediction length, where the former is a hyperparameter and the latter is specific to each dataset. Furthermore, we extract independent sets of subseries for each dataset and set the maximum number of subseries per dataset to 100,000. Subseries are selected uniformly at random without replacement, with frequencies proportional to the lengths of individual time series. In other words, we extract more subseries from longer time series in the training data and fewer from shorter ones. For multivariate time series, we extract independent sets of subseries corresponding to each output. To obtain a fair comparison to GP baselines, we optimize kernel hyperparameters, such as lengthscales, amplitudes, and spectral mixture weights, on the same set of subseries which is used as historical data for Empirical GP.

Following Aksu et al. (2024), we evaluate models by ranking them based on their continuous ranked probability score (CRPS) relative to the Seasonal Naive baseline. Table 1 reports the mean and standard error of average ranks across

models, per domain and overall. Notably, the Empirical GP achieves the highest overall average rank across the category of statistical models, and surprisingly outperforms four out of the eight deep learning models. Crucially, the ranks reported in Table 1 are computed jointly across all models simultaneously, adhering to the standard GIFT-Eval evaluation protocol (Aksu et al., 2024). This allows a direct comparison between the statistical and deep learning blocks.

We attribute this competitiveness to the Empirical GP's explicit modeling of the autocorrelation patterns that dominate time series data. Conversely, deep learning (DL) baselines trained from scratch must implicitly learn these dependencies, often resulting in sample inefficiency and optimization instability. While massive pre-trained foundation models define the state-of-the-art, our results highlight that the Empirical GP remains surprisingly competitive to canonical DL baselines, *trained on the same data*, despite requiring no gradient-based optimization—offering a robust and exceedingly simple complement to complex neural architectures.

Figure 5 illustrates the effects of varying the context length and the number of historical subseries. Without SVD acceleration from Section 3.5, the runtime of the Empirical GP increases significantly as the context length increases. Using SVD acceleration results in virtually negligible runtime. Additionally, increasing the context length improves the performance of the Empirical GP. However, given a fixed amount of training data, increasing the context length implies fewer historical subseries, resulting in a trade-off. In terms of the number of historical subseries, Figure 5 shows that SVD acceleration leads to consistent runtime improvements across various numbers of subseries, and that performance tends to improve and become more stable as the amount of historical data increases.

### 4.4. Learning Curve Extrapolation on LCBench

To demonstrate that Empirical GPs can also leverage sparsely observed historical data, we next consider the task of learning curve extrapolation. This may naturally feature sparsely observed historical data, e.g., due to infrequent logging or early stopping of jobs. We simulate this setting using data from LCBench (Zimmer et al., 2021), which provides learning curve data for neural networks trained for 50 epochs across 35 different datasets with 2,000 randomly sampled hyperparameter configurations per dataset.

For each dataset, we extrapolate partially observed learning curves on a test split with 1,000 examples. From the remaining curves, we construct historical data for the Empirical GP. To simulate sparse historical data due to early stopping, we provide 60% of historical curves as fully observed and truncate the remaining 40% uniformly at random between 20% to 80% of the total number of epochs.

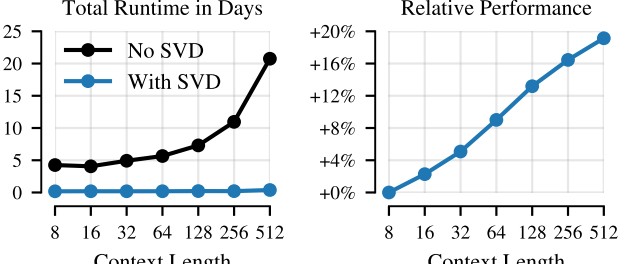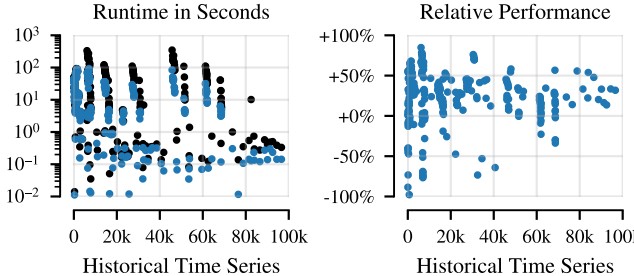

*Figure 5.* Runtime and performance statistics of Empirical GP on the GIFT-Eval time series forecasting benchmark. **Left:** As the context length increases, SVD acceleration significantly decreases runtime to a virtually negligible amount. **Left middle:** Increasing the context length consistently improves the performance of Empirical GP relative to itself with a shorter context length. **Right middle:** SVD acceleration delivers consistent runtime improvements for varying amounts of historical data. **Right:** Increasing the amount of historical data tends to improve and stabilize the performance of Empirical GP relative to the Seasonal Naive baseline.

*Table 1.* Average ranks (mean ± standard error) of various models on the GIFT-Eval time series forecasting benchmark, by domain and overall. Empirical GP (ours) achieves the highest rank among statistical models, and outperforms four out of eight deep learning baselines

|  | Model | Econ/Fin | Energy | Healthcare | Nature | Sales | Transport | Web/CloudOps | Overall |
|---|---|---|---|---|---|---|---|---|---|
| **Statistical Models** | Naive | 10.17 ± 0.80 | 12.75 ± 0.54 | 12.40 ± 1.19 | 14.53 ± 0.38 | 14.75 ± 0.22 | 15.27 ± 0.18 | 11.65 ± 0.57 | 13.09 ± 0.28 |
| | Gaussian RBF GP | 14.83 ± 0.15 | 11.97 ± 0.68 | 13.80 ± 0.33 | 10.73 ± 0.57 | 10.75 ± 0.65 | 12.20 ± 0.40 | 13.55 ± 0.60 | 12.36 ± 0.30 |
| | Auto ETS | 5.67 ± 1.37 | 10.59 ± 0.86 | 4.80 ± 1.53 | 11.93 ± 1.13 | 12.50 ± 1.15 | 13.07 ± 0.57 | 11.75 ± 0.76 | 10.90 ± 0.46 |
| | Spectral Mixture GP | 13.17 ± 0.60 | 10.53 ± 0.63 | 14.00 ± 0.63 | 9.20 ± 0.76 | 9.50 ± 1.35 | 9.20 ± 0.65 | 12.70 ± 0.75 | 10.87 ± 0.35 |
| | Auto Theta | 4.50 ± 0.87 | 11.75 ± 0.72 | 8.00 ± 1.33 | 11.73 ± 0.75 | 9.00 ± 1.27 | 13.47 ± 0.50 | 8.15 ± 1.11 | 10.52 ± 0.45 |
| | Seasonal Naive | 7.50 ± 1.17 | 7.88 ± 0.43 | 9.40 ± 1.34 | 11.93 ± 0.37 | 13.25 ± 0.41 | 11.27 ± 0.43 | 9.70 ± 0.90 | 9.68 ± 0.32 |
| | Auto Arima | 3.50 ± 0.84 | 7.53 ± 0.49 | 4.60 ± 1.40 | 9.67 ± 0.91 | 8.25 ± 1.14 | 10.93 ± 0.46 | 10.45 ± 0.81 | 8.62 ± 0.37 |
| | Empirical GP (ours) | 8.67 ± 1.26 | 7.72 ± 0.85 | 5.20 ± 1.68 | 7.73 ± 0.60 | 8.50 ± 1.44 | 5.87 ± 0.32 | 8.50 ± 1.18 | **7.56 ± 0.42** |
| **Deep Learning** | Crossformer | 16.00 ± 0.00 | 9.62 ± 0.87 | 11.60 ± 2.66 | 7.60 ± 1.54 | 15.75 ± 0.22 | 8.00 ± 1.85 | 7.75 ± 0.64 | 9.42 ± 0.56 |
| | DLinear | 11.67 ± 0.45 | 9.03 ± 0.37 | 10.00 ± 1.26 | 8.80 ± 0.67 | 8.75 ± 0.74 | 9.53 ± 0.46 | 8.80 ± 0.63 | 9.23 ± 0.24 |
| | N-BEATS | 7.83 ± 1.12 | 9.19 ± 0.50 | 9.20 ± 0.95 | 8.73 ± 0.69 | 6.50 ± 1.30 | 7.67 ± 0.36 | 6.90 ± 0.52 | 8.22 ± 0.27 |
| | DeepAR | 9.83 ± 1.52 | 9.56 ± 0.82 | 6.80 ± 1.86 | 6.93 ± 1.27 | 3.00 ± 0.35 | 4.40 ± 0.83 | 8.10 ± 1.05 | 7.66 ± 0.49 |
| | TiDE | 11.33 ± 0.84 | 6.16 ± 0.66 | 9.60 ± 1.82 | 7.80 ± 1.03 | 8.25 ± 1.43 | 5.80 ± 0.51 | 7.00 ± 0.60 | 7.11 ± 0.36 |
| | TFT | 3.83 ± 0.72 | 4.22 ± 0.46 | 5.60 ± 1.66 | 3.00 ± 0.42 | 3.25 ± 0.74 | 2.53 ± 0.26 | 4.65 ± 0.78 | 3.87 ± 0.27 |
| | iTransformer | 4.17 ± 0.96 | 4.34 ± 0.73 | 6.20 ± 1.82 | 2.93 ± 0.68 | 2.25 ± 0.54 | 3.40 ± 0.35 | 3.35 ± 0.65 | 3.77 ± 0.33 |
| | PatchTST | 3.33 ± 0.61 | 3.16 ± 0.46 | 4.80 ± 1.11 | 2.73 ± 0.32 | 1.75 ± 0.41 | 3.40 ± 0.32 | 3.00 ± 0.43 | **3.13 ± 0.21** |

We compare Empirical GPs against extrapolation via power laws and LC-PFNs (Adriaensen et al., 2023), a foundation model specifically pre-trained for learning curve extrapolation. In addition, we include the naive baseline of predicting the last observed value, which converges to the ground truth as the observed fraction approaches 100%.

Figure 6 shows results on the validation set in terms of RMSE and CRPS. The Empirical GP clearly outperforms all baselines both in terms of mean prediction and uncertainty calibration, especially in the regime when only a small fraction of the learning curve has been observed.

Figure 7 ablates the performance of the Empirical GP as a function of both the number of historical learning curves available and their completeness (proportion that are fully observed). As expected, greater completeness leads to better prediction quality. Further, we observe approximate log-linear improvement in average percentile rank as a function of the number of historical curves.

**Comparison to GP Meta-Learning Baselines**   To further evaluate the effectiveness of Empirical GPs against frame-

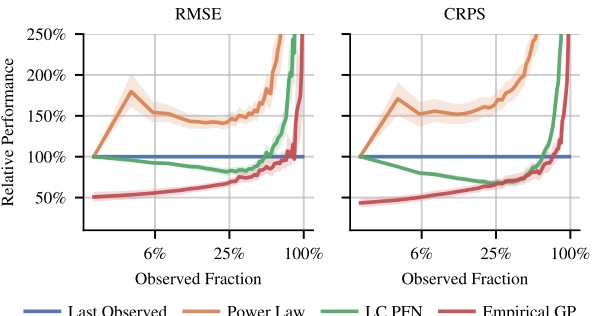

*Figure 6.* Performance metrics, RMSE (left) and CRPS (right), relative to the "last observed" baseline (lower is better), aggregated across all datasets, on the learning curve extrapolation problem.

works explicitly designed to optimize across task corpuses, we benchmark our method against HyperBO (Wang et al., 2024) and PACOH (Rothfuss et al., 2021) (using both its MAP and SVGD variants) on the 1D learning curve extrapolation task. The detailed tabular statistics tracking predictive accuracy (RMSE), uncertainty calibration (CRPS), exact

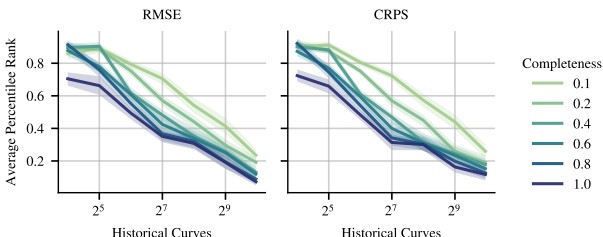

*Figure 7.* Percentile ranks of performance metrics RMSE (left) and CRPS (right) at 40% observed, aggregated across all datasets, on the LCBench learning curve extrapolation problem.

dataset win counts, and runtime profiles across multiple observation splits are provided in Appendix C.

Our non-parametric estimator demonstrates substantial benefits over parametric architectures across these tasks. Specifically, Empirical GP scores the highest total dataset win count across the benchmark, achieving top-tier accuracy on 60 datasets compared to 39 for HyperBO and 18 for PACOH-SVGD. Furthermore, our model maintains the lowest overall mean CRPS error profile, confirming that the data-driven empirical prior produces well-calibrated posterior distributions. Mechanically, because our EM loop relies on stable, closed-form updates, it completely avoids the complex, non-convex optimization landscapes of deep neural kernels. This translates to exceptional speedups, allowing our model to achieve full convergence over $100\times$ faster than particle-based methods like PACOH-SVGD.

### 4.5. 7D Machine Learning Hyperparameter Modeling

To evaluate how Empirical GPs scale to higher-dimensional input spaces where uniform grid discretization is no longer tractable, we evaluate our continuous-domain EM-EGP variant on a 7-dimensional hyperparameter optimization task from LCBench (Zimmer et al., 2021). Here, the model maps 7 continuous black-box hyperparameter dimensions of a neural network directly to its final validation accuracy.

Instead of configuring a rigid coordinate grid, the reference anchor points $\mathbf{Z}$ are chosen as a random subset ($M = 100$) of historically observed input configurations across 30 meta-training datasets. We evaluate downstream extrapolation performance on 5 held-out evaluation datasets across varying local training sample sizes $n_{\text{train}} \in \{5, 10, 20, 50, 100\}$, using $n_{\text{test}} = 50$ configurations for validation evaluation.

As compiled in Table 2, EM-EGP scales robustly to higher-dimensional parameter spaces without requiring structured input alignments. It provides significantly better-calibrated predictive distributions, yielding drastically lower Negative Log-Likelihood (NLL) values across all training sizes compared to alternative methods. This performance is achieved with highly efficient compute overhead, requiring only 9.3 seconds for meta-prior training on standard

*Table 2.* Negative Log-Likelihood (NLL) on 7D LCBench.

| $n_{\text{train}}$ | Empirical GP | HyperBO | Pretrained GP | Vanilla GP |
|---|---|---|---|---|
| 5 | **-0.039** | 0.628 | 0.458 | 0.651 |
| 10 | **-0.150** | 0.547 | 0.155 | 0.447 |
| 20 | **-0.232** | 0.497 | 0.166 | 2.137 |
| 50 | **-0.361** | 0.442 | -0.084 | 0.469 |
| 100 | **-0.415** | 0.418 | -0.172 | 0.042 |

CPUs. Due to space constraints, the corresponding Root Mean Squared Error (RMSE) accuracy metrics – where EM-EGP demonstrates strong data efficiency in the sparse-data regime ($n_{\text{train}} \leq 20$) – are deferred to Appendix C.

## 5. Conclusion

We studied Empirical Gaussian Processes, a principled framework for learning non-parametric GP priors from related datasets. Our work advances both the theoretical understanding and practical applicability of this approach.

We provide a theoretical foundation, proving that Empirical GPs converge to the best Gaussian approximation of the true data-generating process. Our method provides closed-form updates that avoid compute-intensive neural-network based techniques such as HyperBO and Deep Kernel Learning. In the general case of non-shared inputs, the EM algorithm typically converges in a few iterations and scales linearly in the number of data sets, enabling efficient learning from large collections of historical data. For dense observations, we achieve strong modeling performance through a simple algorithm without EM-steps. These results hold even using interpolation to deal with non-uniform observations.

Our experiments on time series forecasting and learning curve extrapolation demonstrate that Empirical GPs achieve performance that is competitive with multiple recent deep learning baselines on practically highly relevant problems while requiring orders of magnitude less training time.

**Limitations and Future Work** Empirical GPs target a middle ground between data-scarce regimes and large-corpus regimes where foundation models are more expressive. With few historical realizations, the empirical covariance is high-variance and can be singular, possibly requiring shrinkage regularization such as an Inverse-Wishart prior. The dense interpolation variant (Section 3.4) needs a reference grid and is thus limited to low dimensions. Leveraging a random subset of historical inputs as reference points is effective (Section C). Learning $\mathbf{Z}$ via $k$-means or Type-II MLE is left to future work. EM only recovers the best *Gaussian* fit under finite second moments; heavy-tailed or non-Gaussian structure, which structured kernels (e.g., Gibbs) or non-Gaussian processes can target, is beyond our scope.

## Acknowledgements

We thank Samuel Müller for helpful discussions and insights about the GIFT-Eval benchmark.

## Impact Statement

This paper presents work whose goal is to advance the field of Machine Learning. There are many potential societal consequences of our work, none which we feel must be specifically highlighted here.

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

# A. Theory

## A.1. Function-Space View

In this section, we provide derivations for our theoretical results from Section 3.

Let $(\mathcal{X}, \rho)$ be a compact metric space. Let $C(\mathcal{X})$ denote the separable Banach space of continuous real-valued functions on $\mathcal{X}$, equipped with the supremum norm $\|h\|_\infty = \sup_{\mathbf{x} \in \mathcal{X}} |h(\mathbf{x})|$ for all $h \in C(\mathcal{X})$. Let $f$ be drawn from a stochastic process indexed by $\mathcal{X}$ with mean function $m(\mathbf{x}) = \mathbb{E}[f(\mathbf{x})]$ and covariance function $k(\mathbf{x}, \mathbf{x}') = \mathrm{Cov}(f(\mathbf{x}), f(\mathbf{x}'))$. We assume that $f$ has almost surely continuous sample paths on $\mathcal{X}$ and finite second moments $\mathbb{E}[\|f\|_\infty^2] < \infty$. Note that, the latter is implied by Fernique's theorem (Fernique, 1970) if $f$ follows a *Gaussian* process. Let $f_1, ..., f_S$ be $S$ i.i.d. sample paths of $f$. We define the *empirical* mean function $m_S$ and the *empirical* covariance function $k_S$ as

$$m_S(\mathbf{x}) = \frac{1}{S} \sum_{i=1}^{S} f_i(\mathbf{x}), \qquad k_S(\mathbf{x}, \mathbf{x}') = \frac{1}{S} \sum_{i=1}^{S} (f_i(\mathbf{x}) - m_S(\mathbf{x}))(f_i(\mathbf{x}') - m_S(\mathbf{x}')).$$

We first show that the empirical mean and covariance functions almost surely converge uniformly to their true counterparts.

**Lemma 1.** *Suppose that $f$ has almost surely continuous sample paths on $\mathcal{X}$ and finite second moments $\mathbb{E}[\|f\|_\infty^2] < \infty$. Note that, the latter is implied by Fernique's theorem (Fernique, 1970) if $f$ is a* Gaussian *process. In the limit of $S \to \infty$, we have*

$$\|m_S - m\|_\infty \xrightarrow{\text{a.s.}} 0 \quad \text{and} \quad \|k_S - k\|_\infty \xrightarrow{\text{a.s.}} 0,$$

*that is, the empirical mean and covariance function almost surely converge uniformly to their true counterparts.*

*Proof.* First, note that the sample paths $f_1, ..., f_S$ are i.i.d. random elements taking values in the separable Banach space $C(\mathcal{X})$. Applying Mourier's strong law of large numbers (Mourier, 1953), we obtain

$$\left\| \frac{1}{S} \sum_{i=1}^{S} f_i - \mathbb{E}[f] \right\|_\infty \xrightarrow{\text{a.s.}} 0 \quad \text{in } C(\mathcal{X}).$$

Since $\mathbb{E}[f(\mathbf{x})] = m(\mathbf{x})$, we have $\|m_S - m\|_\infty \xrightarrow{\text{a.s.}} 0$.

For the empirical covariance function, we consider the random element $h = f \otimes f$, defined by $h(\mathbf{x}, \mathbf{x}') = f(\mathbf{x})f(\mathbf{x}')$, where $f \in C(\mathcal{X})$ implies $h \in C(\mathcal{X} \times \mathcal{X})$. Additionally, since $\mathcal{X}$ is a compact metric space, $\mathcal{X} \times \mathcal{X}$ is also a compact metric space, implying that $C(\mathcal{X} \times \mathcal{X})$ is a separable Banach space with supremum norm

$$\|h\|_\infty = \sup_{(\mathbf{x}, \mathbf{x}') \in \mathcal{X} \times \mathcal{X}} |h(\mathbf{x}, \mathbf{x}')| = \|f\|_\infty^2.$$

Since $\mathbb{E}[\|f\|_\infty^2] < \infty$, we have $\mathbb{E}[\|h\|_\infty] < \infty$. By applying Mourier's strong law of large numbers (Mourier, 1953) again, we obtain

$$\left\| \frac{1}{S} \sum_{i=1}^{S} h_i - \mathbb{E}[h] \right\|_\infty \xrightarrow{\text{a.s.}} 0 \quad \text{in } C(\mathcal{X} \times \mathcal{X}),$$

with $\mathbb{E}[h(\mathbf{x}, \mathbf{x}')] = \mathbb{E}[f(\mathbf{x})f(\mathbf{x}')] = k(\mathbf{x}, \mathbf{x}') + m(\mathbf{x})m(\mathbf{x}')$.

Since the mapping $(u, v) \mapsto u \otimes v$ from $C(\mathcal{X}) \times C(\mathcal{X}) \to C(\mathcal{X} \times \mathcal{X})$ is continuous, $\|m_S - m\|_\infty \xrightarrow{\text{a.s.}} 0$ implies

$$m_S(\mathbf{x})m_S(\mathbf{x}') \xrightarrow{\text{a.s.}} m(\mathbf{x})m(\mathbf{x}') \quad \text{uniformly on } \mathcal{X} \times \mathcal{X},$$

via the continuous mapping theorem. Combining the two results above with the continuity of subtraction yields

$$k_S \xrightarrow{\text{a.s.}} (k + m \otimes m) - (m \otimes m) = k \quad \text{uniformly on } \mathcal{X} \times \mathcal{X},$$

which gives the claim. $\qquad\square$

For any finite set of points $\mathbf{x}_1, ..., \mathbf{x}_N \in \mathcal{X}$, we define

$$\mathbf{m} = \left[m(\mathbf{x}_i)\right]_{i=1}^N, \qquad \mathbf{m}_S = \left[m_S(\mathbf{x}_i)\right]_{i=1}^N, \qquad \mathbf{K} = \left[k(\mathbf{x}_i, \mathbf{x}_j)\right]_{i,j=1}^N, \qquad \mathbf{K}_S = \left[k_S(\mathbf{x}_i, \mathbf{x}_j)\right]_{i,j=1}^N.$$

Note that $k_S$ is a valid covariance function, because $\mathbf{K}$ is positive semi-definite, that is, for any $\mathbf{u} \in \mathbb{R}^n$

$$\mathbf{u}^\mathsf{T} \mathbf{K}_S \mathbf{u} = \frac{1}{S} \sum_{i=1}^S \left( \sum_{j=1}^N u_j (f_i(\mathbf{x}_j) - m_S(\mathbf{x}_j)) \right)^2 \geq 0.$$

Next, we show that the finite-dimensional multivariate normal random variable $\mathbf{g}_S \sim \mathcal{N}(\mathbf{m}_S, \mathbf{K}_S)$, induced by the empirical mean and covariance functions, converges in distribution to $\mathbf{g} \sim \mathcal{N}(\mathbf{m}, \mathbf{K})$, the corresponding multivariate normal random variable induced by the true mean and covariance functions.

**Lemma 2.** *For almost every sequence of sample paths $\{f_i\}_{i=1}^S$ and any finite set of points $\mathbf{x}_1, ..., \mathbf{x}_N \in \mathcal{X}$,*

$$\mathbf{g}_S \xrightarrow{d} \mathbf{g} \quad as \quad S \to \infty,$$

*that is, the multivariate normal random variable induced by the empirical mean and covariance functions converges in distribution to the corresponding multivariate normal random variable induced by the true mean and covariance function.*

*Proof.* The characteristic function of $\mathbf{g}_S \sim \mathcal{N}(\mathbf{m}_S, \mathbf{K}_S)$ is given by

$$\varphi_S(\mathbf{t}) = \exp\left( i\mathbf{t}^\mathsf{T} \mathbf{m}_S - \frac{1}{2}\mathbf{t}^\mathsf{T} \mathbf{K}_S \mathbf{t} \right), \quad \forall \mathbf{t} \in \mathbb{R}^N.$$

Applying Lemma 1, we obtain element-wise convergence of $\mathbf{m}_S \to \mathbf{m}$ and $\mathbf{K}_S \to \mathbf{K}$ for almost every realization of the sample paths. By continuity of $\varphi_S$, we obtain pointwise convergence of

$$\varphi_S(\mathbf{t}) \to \varphi(\mathbf{t}) = \exp\left( i\mathbf{t}^\mathsf{T} \mathbf{m} - \frac{1}{2}\mathbf{t}^\mathsf{T} \mathbf{K} \mathbf{t} \right), \quad \forall \mathbf{t} \in \mathbb{R}^N,$$

for almost every realization of $\{f_i\}_{i=1}^S$, as $S \to \infty$. Observing that $\varphi$ is the characteristic function of $\mathbf{g} \sim \mathcal{N}(\mathbf{m}, \mathbf{K})$ and continuous at $\mathbf{t} = \mathbf{0}$, we apply Lévy's continuity theorem (Lévy, 1925) and conclude the claim. $\square$

Let $\mathcal{GP}(m_S, k_S)$ be a GP defined by the empirical mean function $m_S$ and empirical covariance function $k_S$, conditioned on observed sample paths $f_1, ..., f_S$. We refer to $\mathcal{GP}(m_S, k_S)$ as Empirical GP. Additionally, let $\mathcal{GP}(m, k)$ be a Gaussian process defined by the true mean function $m$ and true covariance function $k$. We refer to $\mathcal{GP}(m, k)$ as *limiting* GP.

**Proposition 1.** *Assume that $k$ is continuous and that its canonical semi-metric satisfies Dudley's entropy integral condition. Then, for almost every sequence of sample paths $\{f_i\}_{i=1}^S$, we have $\mathcal{GP}(m_S, k_S) \rightharpoonup \mathcal{GP}(m, k)$ as $S \to \infty$.*

*Proof.* To establish the weak convergence of $\mathcal{GP}(m_S, k_S) \rightharpoonup \mathcal{GP}(m, k)$ in $C(\mathcal{X})$, we must verify the convergence of finite-dimensional distributions and the tightness of the sequence of measures induced by $\{\mathcal{GP}(m_S, k_S)\}_{S=1}^\infty$. The convergence of finite-dimensional distributions is established directly by Lemma 2. For tightness in $C(\mathcal{X})$, Dudley's condition implies $\mathcal{GP}(m, k)$ is supported on $C(\mathcal{X})$. Uniform convergence $k_S \to k$, established by Lemma 1, implies $d_{k_S} \to d_k$ uniformly, and thus the entropy integrals of $(\mathcal{X}, d_{k_S})$ are uniformly bounded for all sufficiently large $S$. By Dudley's theorem (Dudley, 1967), $\{\mathcal{GP}(m_S, k_S)\}_{S=1}^\infty$ is tight in $C(\mathcal{X})$. Finally, convergence of finite-dimensional distributions plus tightness yields weak convergence in $C(\mathcal{X})$ via Prokhorov's theorem (Prokhorov, 1956). $\square$

This weak convergence implies weak convergence of posteriors under Bayesian inference.

**Corollary 1.** *Given a dataset $\mathcal{D}$, for almost every sequence of sample paths $\{f_i\}_{i=1}^S$, $\mathcal{GP}(m_S, k_S) \,|\, \mathcal{D} \rightharpoonup \mathcal{GP}(m, k) \,|\, \mathcal{D}$ as $S \to \infty$, that is, the posterior of $\mathcal{GP}(m_S, k_S)$ conditioned on $\mathcal{D}$ converges weakly to the posterior of $\mathcal{GP}(m, k)$ in $C(X)$.*

*Proof.* The posterior mean and covariance functions are continuous mappings of the corresponding prior mean and covariance functions. Therefore, they preserve the uniform convergence properties established by Lemma 1. An argument analogous to Proposition 1 gives the claim. $\square$

Finally, we show that the limiting Gaussian process $g$ is the best possible Gaussian approximation, in terms of KL divergence, of the true underlying stochastic process $f$.

**Proposition 2.** *Let $\mathbb{P}$ denote the law of $f$, and let $\mathbb{G}$ be any GP indexed by $\mathcal{X}$. Assume that $\mathbb{P}$ is absolutely continuous with respect to $\mathbb{G}$, and that $k$ is strictly positive definite. Then, $\lim_{S \to \infty} \mathcal{GP}(m_S, k_S) = \operatorname{argmin}_{\mathbb{G}} D_{\mathrm{KL}}(\mathbb{P} \parallel \mathbb{G})$.*

*Proof.* Following (Sun et al., 2019), we define the KL divergence between stochastic processes via finite-dimensional marginals,

$$D_{\mathrm{KL}}(\mathbb{P} \parallel \mathbb{G}) = \sup_{\mathbf{X} \subset \mathcal{X},\ |\mathbf{X}| < \infty} D_{\mathrm{KL}}(\mathbb{P}_{\mathbf{f_X}} \parallel \mathbb{G}_{\mathbf{h_X}}).$$

For any finite $\mathbf{X} \subset \mathcal{X}$, let $p_{\mathbf{X}}$ denote the distribution of $\mathbf{f_X}$ with mean $\boldsymbol{\mu_X}$ and covariance $\boldsymbol{\Sigma_X}$. For a GP $\mathbb{G}$ with mean $\tilde{m}$ and covariance $\tilde{k}$, we have $\mathbf{h_X} \sim \mathcal{N}(\tilde{\boldsymbol{\mu}}_{\mathbf{X}}, \tilde{\boldsymbol{\Sigma}}_{\mathbf{X}})$. Decomposing the KL divergence into entropy and cross-entropy, we have

$$D_{\mathrm{KL}}(p_{\mathbf{X}} \parallel \mathcal{N}(\tilde{\boldsymbol{\mu}}_{\mathbf{X}}, \tilde{\boldsymbol{\Sigma}}_{\mathbf{X}})) = -H(p_{\mathbf{X}}) + H(p_{\mathbf{X}}, \mathcal{N}(\tilde{\boldsymbol{\mu}}_{\mathbf{X}}, \tilde{\boldsymbol{\Sigma}}_{\mathbf{X}})).$$

Since the entropy $H(p_{\mathbf{X}})$ is independent of $\mathbb{G}$, minimizing the KL divergence is equivalent to minimizing the cross-entropy. For a Gaussian target, the cross-entropy depends on $p_{\mathbf{X}}$ only through its first two moments,

$$H(p_{\mathbf{X}}, \mathcal{N}(\tilde{\boldsymbol{\mu}}_{\mathbf{X}}, \tilde{\boldsymbol{\Sigma}}_{\mathbf{X}})) = \frac{1}{2} \log |2\pi \tilde{\boldsymbol{\Sigma}}_{\mathbf{X}}| + \frac{1}{2} \operatorname{tr}(\tilde{\boldsymbol{\Sigma}}_{\mathbf{X}}^{-1} \boldsymbol{\Sigma}_{\mathbf{X}}) + \frac{1}{2}(\boldsymbol{\mu}_{\mathbf{X}} - \tilde{\boldsymbol{\mu}}_{\mathbf{X}})^{\top} \tilde{\boldsymbol{\Sigma}}_{\mathbf{X}}^{-1} (\boldsymbol{\mu}_{\mathbf{X}} - \tilde{\boldsymbol{\mu}}_{\mathbf{X}}).$$

The cross-entropy is strictly convex in the natural parameters of the Gaussian, and is uniquely minimized when $\tilde{\boldsymbol{\mu}}_{\mathbf{X}} = \boldsymbol{\mu}_{\mathbf{X}}$ and $\tilde{\boldsymbol{\Sigma}}_{\mathbf{X}} = \boldsymbol{\Sigma}_{\mathbf{X}}$.

By Proposition 1, the mean and covariance of $\lim_{S \to \infty} \mathcal{GP}(m_S, k_S)$ converge to their true counterparts, such that, for every finite $\mathbf{X}$, the marginal $\mathbf{g_X} \sim \mathcal{N}(\boldsymbol{\mu}_{\mathbf{X}}, \boldsymbol{\Sigma}_{\mathbf{X}})$ uniquely minimizes the marginal KL divergence. Any other GP differs on some finite $\mathbf{X}^*$, yielding a strictly larger KL divergence on that marginal, and hence a strictly larger supremum. $\square$

### A.2. EM-based Model

**Lemma 3.** *[Shift-Interpolant] Let the interpolated covariance at new input locations $\mathbf{X}_*$ be defined as:*

$$\boldsymbol{\Sigma}(\mathbf{X}_*, \mathbf{X}_*) = k(\mathbf{X}_*, \mathbf{X}_*) + \mathbf{W} \boldsymbol{\delta}_{\Sigma} \mathbf{W}^{\top}, \tag{3}$$

*where $\mathbf{W} = \mathbf{K}_{*X} \mathbf{K}_{XX}^{-1}$ and $\boldsymbol{\delta}_{\Sigma} = \boldsymbol{\Sigma} - \mathbf{K}_{XX}$. If the EM-optimized covariance $\boldsymbol{\Sigma}$ is positive semi-definite (PSD), and the base kernel $k(\cdot, \cdot)$ is a valid covariance function that is positive (semi-)definite, then $\boldsymbol{\Sigma}(\mathbf{X}_*, \mathbf{X}_*)$ is positive (semi-)definite.*

*Proof.* We begin by expanding the definition of the interpolated covariance. Let $\mathbf{K}_{**} = k(\mathbf{X}_*, \mathbf{X}_*)$, $\mathbf{K}_{*X} = k(\mathbf{X}_*, \mathbf{X})$, and $\mathbf{K}_{XX} = k(\mathbf{X}, \mathbf{X})$. Substituting the definition of the residual $\boldsymbol{\delta}_{\Sigma}$ into the interpolation formula:

$$\begin{aligned}
\boldsymbol{\Sigma}(\mathbf{X}_*, \mathbf{X}_*) &= \mathbf{K}_{**} + \mathbf{W}(\boldsymbol{\Sigma} - \mathbf{K}_{XX})\mathbf{W}^{\top} \\
&= \mathbf{K}_{**} + \mathbf{W}\boldsymbol{\Sigma}\mathbf{W}^{\top} - \mathbf{W}\mathbf{K}_{XX}\mathbf{W}^{\top}.
\end{aligned}$$

We simplify the term $\mathbf{W}\mathbf{K}_{XX}\mathbf{W}^{\top}$ using the definition of the weight matrix $\mathbf{W} = \mathbf{K}_{*X}\mathbf{K}_{XX}^{-1}$:

$$\begin{aligned}
\mathbf{W}\mathbf{K}_{XX}\mathbf{W}^{\top} &= (\mathbf{K}_{*X}\mathbf{K}_{XX}^{-1})\mathbf{K}_{XX}(\mathbf{K}_{*X}\mathbf{K}_{XX}^{-1})^{\top} \\
&= \mathbf{K}_{*X}(\mathbf{K}_{XX}^{-1}\mathbf{K}_{XX})\mathbf{K}_{XX}^{-1}\mathbf{K}_{X*} \\
&= \mathbf{K}_{*X}\mathbf{K}_{XX}^{-1}\mathbf{K}_{X*}.
\end{aligned}$$

Substituting this back into the expansion above, we can rearrange the interpolated covariance as the sum of two distinct matrices, denoted $\mathbf{A}$ and $\mathbf{B}$:

$$\boldsymbol{\Sigma}(\mathbf{X}_*, \mathbf{X}_*) = \underbrace{\left(\mathbf{K}_{**} - \mathbf{K}_{*X}\mathbf{K}_{XX}^{-1}\mathbf{K}_{X*}\right)}_{\mathbf{A}} + \underbrace{\left(\mathbf{W}\boldsymbol{\Sigma}\mathbf{W}^{\top}\right)}_{\mathbf{B}}.$$

To prove that $\boldsymbol{\Sigma}(\mathbf{X}_*, \mathbf{X}_*) \succeq 0$, it suffices to show that both $\mathbf{A}$ and $\mathbf{B}$ are positive semi-definite.

**1. Analysis of A (Schur Complement)** The matrix $\mathbf{A}$ is recognized as the conditional covariance of the base Gaussian Process at locations $\mathbf{X}_*$ given observations at $\mathbf{X}$. Consider the joint kernel matrix of the union of grid points and new points, which is PSD by the definition of the kernel function $k(\cdot, \cdot)$:

$$\mathbf{K}_{\text{joint}} = \begin{bmatrix} \mathbf{K}_{XX} & \mathbf{K}_{X*} \\ \mathbf{K}_{*X} & \mathbf{K}_{**} \end{bmatrix} \succeq 0.$$

By the properties of the Schur complement, if a block matrix is PSD, the Schur complement of its principal submatrix $\mathbf{K}_{XX}$ is also PSD. Therefore:

$$\mathbf{A} = \mathbf{K}_{**} - \mathbf{K}_{*X}\mathbf{K}_{XX}^{-1}\mathbf{K}_{X*} \succeq 0.$$

**2. Analysis of B (Projected Learned Covariance)** The matrix $\mathbf{B} = \mathbf{W}\boldsymbol{\Sigma}\mathbf{W}^\top$ is a congruence transformation of $\boldsymbol{\Sigma}$. From the M-step of the EM algorithm, $\boldsymbol{\Sigma}$ is updated as a sum of posterior covariances and outer products of mean adjustments:

$$\boldsymbol{\Sigma}^{(t+1)} = \frac{1}{K-1}\sum_{i=1}^{K}\mathbf{C}_i + \tilde{\mathbf{m}}_i\tilde{\mathbf{m}}_i^\top.$$

Since covariance matrices $\mathbf{C}_i$ and outer products $\tilde{\mathbf{m}}_i\tilde{\mathbf{m}}_i^\top$ are always positive semi-definite, their sum $\boldsymbol{\Sigma}$ is PSD ($\boldsymbol{\Sigma} \succeq 0$). For any vector $\mathbf{v}$, let $\mathbf{u} = \mathbf{W}^\top\mathbf{v}$. Then:

$$\mathbf{v}^\top\mathbf{B}\mathbf{v} = \mathbf{v}^\top\mathbf{W}\boldsymbol{\Sigma}\mathbf{W}^\top\mathbf{v} = \mathbf{u}^\top\boldsymbol{\Sigma}\mathbf{u} \geq 0.$$

Thus, $\mathbf{B} \succeq 0$.

**Conclusion** Since $\boldsymbol{\Sigma}(\mathbf{X}_*, \mathbf{X}_*) = \mathbf{A} + \mathbf{B}$ is the sum of two positive semi-definite matrices, it is itself positive semi-definite. Further, if the base kernel $k$ is positive definite (strict inequality), then $\boldsymbol{\Sigma}(\mathbf{X}_*, \mathbf{X}_*) \succeq \mathbf{A} \succ 0$. $\qquad\square$

### A.3. Computational Complexity of the EM-Algorithm With Residual Interpolation

Substituting the dense $\mathbf{Q}_i$ for $\sigma^2\mathbf{I}$ in $\boldsymbol{\Sigma}_{\mathbf{X}_i,\mathbf{X}_i}$ forfeits the low-rank-plus-diagonal structure exploited by the Woodbury identity, raising the E-step inversion from $\mathcal{O}(M^3 + N_iM^2)$ to $\mathcal{O}(N_i^3)$ in the large-observation regime $N_i > M$. However, it is required to make the variance-starvation fix consistent during training on sparse, irregular data.

## B. Additional Details of the Experiments

### B.1. Datasets

GIFT-Eval and LCBench are available under the Apache License 2.0. S&P 500 data on Kaggle was retrieved under the CC0 License. The Mauna Loa dataset is available under the CC0 License.

### B.2. Baseline Implementations

LC-PFN is available on GitHub under the MIT License at `https://github.com/automl/lcpfn`

### B.3. Compute Usage

One benefit of Empirical GPs is that they are quite compute-efficient. All of our experiments were performed on Intel Cooper Lake CPUs, using less than 3,000 CPU hours in total.

### B.4. EM-Algorithm Convergence Speed

Figure 8 shows the $L_2$ norm of the difference of subsequent iterates of the EM-estimated prior mean $\mu^{(t)}$ and covariance $\Sigma^{(t)}$, as a function of the number of noisy independent incomplete datasets (S), and for different numbers of observed data points per dataset (n). Notably, convergence speed to a fixed-point of the EM-iterations depends on the number of observed entries $n$, compared to the size of the full latent vector $N = 128$ in this case. Note that the reason $n = 128$ is not converging within one iteration in this case is because this is studying the noisy case, where $\sigma^2 = 10^{-4}$, showing that even in the full observed case, the EM-iterations with $\sigma^2 > 0$ require a number of iterations to converge, in order to de-noise the latent vectors $\{\mathbf{f}_i\}$.

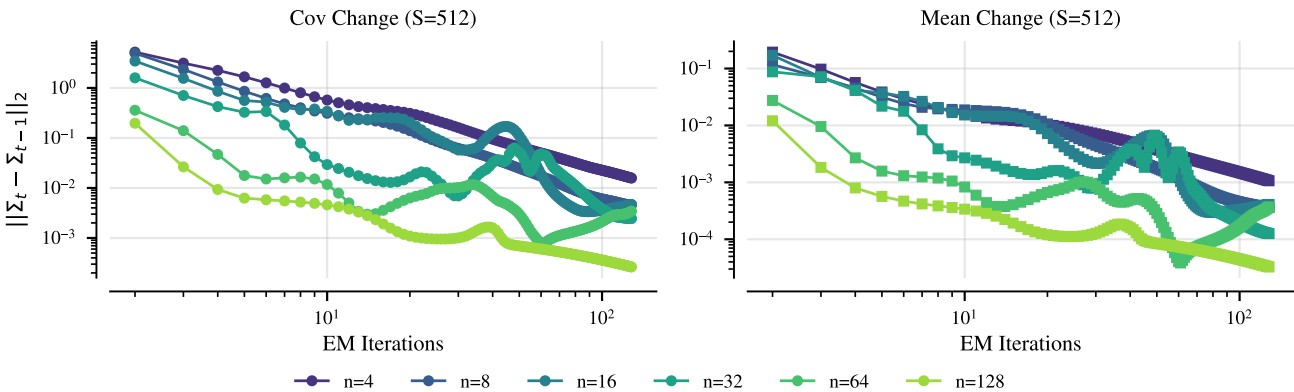

*Figure 8.* $L_2$ norm of the difference of subsequent iterates of the EM-estimated prior mean $\mu^{(t)}$ and covariance $\Sigma^{(t)}$, as a function of the number of independent incomplete datasets (S), and for different numbers of observed data points per dataset (n).

# C. Additional Empirical Evaluations

This section provides comprehensive experimental results to supplement the main text, presenting both the high-dimensional scalability accuracy metrics and the detailed 1D meta-learning baseline comparisons.

## C.1. Additional Learning Curve Extrapolation Results

Table 3 shows the average rank, and Table 4 shows the performance metrics at 40% observed fraction for the learning curve extrapolation benchmark of Section 4.4.

*Table 3.* Average rank for the 1D LCBench learning curve extrapolation benchmark of Section 4.4

| | RMSE | | | | CRPS | | | |
|---|---|---|---|---|---|---|---|---|
| Observed | Last Observed | Power Law | LC PFN | Empirical GP | Last Observed | Power Law | LC PFN | Empirical GP |
| 10% | $2.91 \pm 0.37$ | $3.94 \pm 0.34$ | $2.14 \pm 0.36$ | $1.00 \pm 0.00$ | $3.03 \pm 0.30$ | $3.94 \pm 0.24$ | $2.00 \pm 0.24$ | $1.03 \pm 0.17$ |
| 20% | $2.91 \pm 0.28$ | $4.00 \pm 0.00$ | $2.03 \pm 0.38$ | $1.06 \pm 0.24$ | $3.03 \pm 0.17$ | $3.97 \pm 0.17$ | $1.74 \pm 0.44$ | $1.26 \pm 0.44$ |
| 30% | $2.83 \pm 0.45$ | $3.97 \pm 0.17$ | $1.97 \pm 0.57$ | $1.23 \pm 0.55$ | $2.86 \pm 0.36$ | $4.00 \pm 0.00$ | $1.69 \pm 0.58$ | $1.46 \pm 0.66$ |
| 40% | $2.80 \pm 0.53$ | $3.94 \pm 0.24$ | $2.06 \pm 0.64$ | $1.20 \pm 0.47$ | $2.83 \pm 0.45$ | $4.00 \pm 0.00$ | $1.63 \pm 0.60$ | $1.54 \pm 0.66$ |
| 50% | $2.26 \pm 0.66$ | $3.94 \pm 0.24$ | $2.43 \pm 0.78$ | $1.37 \pm 0.73$ | $2.57 \pm 0.61$ | $4.00 \pm 0.00$ | $1.91 \pm 0.66$ | $1.51 \pm 0.82$ |
| 60% | $2.31 \pm 0.76$ | $3.91 \pm 0.28$ | $2.31 \pm 0.80$ | $1.46 \pm 0.78$ | $2.29 \pm 0.71$ | $4.00 \pm 0.00$ | $2.20 \pm 0.76$ | $1.51 \pm 0.78$ |
| 70% | $2.09 \pm 0.95$ | $3.89 \pm 0.32$ | $2.54 \pm 0.70$ | $1.49 \pm 0.66$ | $1.83 \pm 0.66$ | $4.00 \pm 0.00$ | $2.57 \pm 0.70$ | $1.60 \pm 0.77$ |
| 80% | $1.89 \pm 0.76$ | $3.97 \pm 0.17$ | $2.57 \pm 0.70$ | $1.57 \pm 0.74$ | $1.51 \pm 0.61$ | $4.00 \pm 0.00$ | $2.74 \pm 0.56$ | $1.74 \pm 0.70$ |
| 90% | $1.54 \pm 0.61$ | $4.00 \pm 0.00$ | $2.89 \pm 0.40$ | $1.57 \pm 0.56$ | $1.14 \pm 0.36$ | $4.00 \pm 0.00$ | $2.97 \pm 0.17$ | $1.89 \pm 0.40$ |

## C.2. 7D Hyperparameter Optimization RMSE Results

Table 5 tracks the Root Mean Squared Error (RMSE) performance metrics across the 7-dimensional configuration space of LCBench, serving as a direct accuracy complement to the primary main-text calibration (NLL) findings.

## C.3. 1D Meta-Learning Baseline Comparisons on LCBench

We evaluate the predictive accuracy and uncertainty calibration of our continuous-domain EM Empirical GP variant against HyperBO (Wang et al., 2024) and PACOH (Rothfuss et al., 2021) (both MAP and SVGD variants) across multiple learning curve observed fractions (20%, 40%, 60%, and 80%).

Tables 6 and 7 show the average Root Mean Squared Error (RMSE) and Continuous Ranked Probability Score (CRPS) aggregated across all datasets. Table 8 tracks total dataset win counts (the frequency a given model achieves the top rank), and Table 9 summarizes pre-training and inference wall-clock times.

*Table 4.* Performance metrics at 40% observed fraction for the 1D LCBench learning curve extrapolation benchmark of Section 4.4

| | RMSE | | | | CRPS | | | |
|---|---|---|---|---|---|---|---|---|
| Dataset | Last Observed | Power Law | LC PFN | Empirical GP | Last Observed | Power Law | LC PFN | Empirical GP |
| adult | 3.1908 | 4.6024 | 1.8041 | 4.6075 | 0.0108 | 0.0285 | 0.0063 | 0.0118 |
| airlines | 3.1024 | 3.0310 | 1.9329 | 0.9740 | 0.0182 | 0.0281 | 0.0135 | 0.0092 |
| albert | 1.1373 | 1.6251 | 1.2812 | 1.1045 | 0.0089 | 0.0156 | 0.0082 | 0.0070 |
| Amazon_employee... | 4.5946 | 8.4153 | 4.4209 | 4.5202 | 0.0270 | 0.0629 | 0.0213 | 0.0279 |
| APSFailure | 3.2776 | 6.6912 | 3.0040 | 1.5685 | 0.0103 | 0.0275 | 0.0064 | 0.0057 |
| Australian | 3.9182 | 6.9080 | 3.7916 | 2.6809 | 0.0330 | 0.0628 | 0.0249 | 0.0192 |
| bank-marketing | 2.3401 | 6.2384 | 2.0299 | 2.9862 | 0.0133 | 0.0361 | 0.0088 | 0.0112 |
| blood-transfusi... | 3.4556 | 5.1920 | 4.0090 | 3.2925 | 0.0242 | 0.0453 | 0.0267 | 0.0239 |
| car | 5.3544 | 8.8038 | 4.5520 | 3.4948 | 0.0613 | 0.1054 | 0.0374 | 0.0324 |
| christine | 2.0039 | 2.6694 | 1.4264 | 1.0340 | 0.0147 | 0.0298 | 0.0117 | 0.0078 |
| cnae-9 | 6.4498 | 11.2725 | 4.7778 | 3.4354 | 0.0807 | 0.1396 | 0.0446 | 0.0331 |
| connect-4 | 7.1584 | 7.0396 | 5.4272 | 5.9269 | 0.0429 | 0.0691 | 0.0342 | 0.0555 |
| covertype | 3.6915 | 6.1080 | 3.8519 | 3.5591 | 0.0283 | 0.0565 | 0.0218 | 0.0259 |
| credit-g | 2.2213 | 3.6483 | 2.1637 | 1.9849 | 0.0199 | 0.0331 | 0.0168 | 0.0171 |
| dionis | 4.4809 | 7.1385 | 3.9396 | 2.3296 | 0.1070 | 0.1925 | 0.0735 | 0.0570 |
| fabert | 3.8464 | 5.4327 | 2.8470 | 2.6908 | 0.0609 | 0.0884 | 0.0318 | 0.0307 |
| Fashion-MNIST | 2.5179 | 5.0488 | 2.6076 | 1.5071 | 0.0197 | 0.0415 | 0.0140 | 0.0087 |
| helena | 1.5967 | 2.1588 | 1.4563 | 1.3551 | 0.1274 | 0.1846 | 0.0920 | 0.0881 |
| higgs | 2.1750 | 2.4492 | 1.3215 | 1.3709 | 0.0192 | 0.0287 | 0.0107 | 0.0124 |
| jannis | 3.3169 | 3.9512 | 2.2955 | 2.5233 | 0.0231 | 0.0425 | 0.0163 | 0.0176 |
| jasmine | 2.8734 | 5.2272 | 2.8076 | 2.7209 | 0.0228 | 0.0487 | 0.0165 | 0.0148 |
| jungle_chess_2p... | 3.5999 | 4.3597 | 2.5321 | 3.8737 | 0.0204 | 0.0394 | 0.0133 | 0.0221 |
| kc1 | 5.4086 | 10.3631 | 6.3667 | 4.6170 | 0.0272 | 0.0799 | 0.0300 | 0.0256 |
| KDDCup09_appete... | 4.9707 | 6.9993 | 3.7045 | 3.3654 | 0.0265 | 0.0474 | 0.0187 | 0.0193 |
| kr-vs-kp | 3.2222 | 5.4259 | 2.8233 | 2.8330 | 0.0290 | 0.0510 | 0.0167 | 0.0160 |
| mfeat-factors | 6.9274 | 12.7500 | 5.5009 | 3.8669 | 0.0671 | 0.1406 | 0.0377 | 0.0309 |
| MiniBooNE | 5.0709 | 6.7038 | 3.4769 | 4.1582 | 0.0121 | 0.0309 | 0.0082 | 0.0154 |
| nomao | 2.8692 | 6.1812 | 3.0726 | 1.7033 | 0.0106 | 0.0333 | 0.0081 | 0.0068 |
| numerai28.6 | 0.4795 | 0.4879 | 0.3821 | 0.3305 | 0.0062 | 0.0072 | 0.0042 | 0.0037 |
| phoneme | 2.8501 | 4.9135 | 2.5345 | 2.7296 | 0.0172 | 0.0358 | 0.0135 | 0.0142 |
| segment | 4.7394 | 7.5308 | 5.5354 | 3.6175 | 0.0587 | 0.1071 | 0.0565 | 0.0385 |
| shuttle | 10.7070 | 14.0135 | 9.2256 | 7.8955 | 0.0398 | 0.0880 | 0.0317 | 0.0401 |
| sylvine | 3.0218 | 5.1236 | 2.6685 | 1.9252 | 0.0253 | 0.0467 | 0.0142 | 0.0101 |
| vehicle | 3.5268 | 4.6551 | 3.7274 | 2.8041 | 0.0516 | 0.0675 | 0.0424 | 0.0329 |
| volkert | 2.5433 | 4.0675 | 2.2571 | 2.1124 | 0.0364 | 0.0640 | 0.0251 | 0.0269 |

*Table 5.* RMSE Metrics on 7D LCBench Hyperparameter Space (Lower is Better).

| $n_{train}$ | Empirical GP | HyperBO | Pretrained GP | Vanilla GP | Global Mean |
|---|---|---|---|---|---|
| 5 | **0.244** ±**0.041** | 0.274 ±0.028 | 0.357 ±0.011 | 0.381 ±0.064 | 0.663 ±0.007 |
| 10 | **0.220** ±**0.031** | 0.238 ±0.028 | 0.276 ±0.008 | 0.321 ±0.035 | 0.663 ±0.007 |
| 20 | **0.206** ±**0.028** | 0.215 ±0.009 | 0.273 ±0.028 | 0.283 ±0.024 | 0.663 ±0.007 |
| 50 | 0.186 ±0.026 | **0.181** ±**0.016** | 0.228 ±0.023 | 0.234 ±0.009 | 0.663 ±0.007 |
| 100 | 0.178 ±0.012 | **0.169** ±**0.014** | 0.204 ±0.017 | 0.197 ±0.016 | 0.663 ±0.007 |

*Table 6.* Average RMSE on 1D LCBench (Lower is Better).

| Method | 20% | 40% | 60% | 80% | Overall Mean |
|---|---|---|---|---|---|
| Last Observed | 6.2758 | 3.3719 | 1.9061 | **0.8552** | 3.1023 |
| Empirical GP | **4.4199** | 2.8615 | 1.9226 | 0.9397 | 2.5359 |
| HyperBO | 4.7324 | **2.7482** | **1.6737** | 0.9508 | **2.5263** |
| PACOH-MAP | 8.9320 | 6.3973 | 4.7251 | 3.6093 | 5.9159 |
| PACOH-SVGD | 5.6615 | 3.4254 | 2.3460 | 1.6228 | 3.2639 |

*Table 7.* Average CRPS on 1D LCBench (Lower is Better).

| Method | 20% | 40% | 60% | 80% | Overall Mean |
|---|---|---|---|---|---|
| Last Observed | 0.0623 | 0.0281 | 0.0123 | **0.0040** | 0.0267 |
| Empirical GP | **0.0408** | **0.0207** | **0.0115** | 0.0056 | **0.0196** |
| HyperBO | 0.0462 | 0.0246 | 0.0161 | 0.0122 | 0.0248 |
| PACOH-MAP | 0.0993 | 0.0714 | 0.0555 | 0.0459 | 0.0680 |
| PACOH-SVGD | 0.0605 | 0.0329 | 0.0228 | 0.0180 | 0.0336 |

*Table 8.* Win Counts (Rank = 1) Across LCBench Datasets.

| Method | 20% | 40% | 60% | 80% | Total Wins |
|---|---|---|---|---|---|
| Last Observed | 0 | 3 | 3 | **16** | 22 |
| Empirical GP | **25** | **15** | **13** | 7 | **60** |
| HyperBO | 7 | 11 | 13 | 8 | 39 |
| PACOH-MAP | 0 | 0 | 0 | 1 | 1 |
| PACOH-SVGD | 3 | 6 | 6 | 3 | 18 |

*Table 9.* Pre-training and Inference Run-time (Seconds) Comparisons.

| Method | Pretrain Time (s) | Inference Time (s) | Total Time (s) | Speedup vs. SVGD |
|---|---|---|---|---|
| Empirical GP | **14.35** | 4.38 | **18.72** | **102.3×** |
| HyperBO | 96.75 | **10.74** | 107.49 | 17.8× |
| PACOH-MAP | 287.49 | 14.00 | 301.49 | 6.4× |
| PACOH-SVGD | 1902.67 | 13.80 | 1916.46 | 1.0× |

