# OpenReview forum: "Empirical Gaussian Processes"
_ICML.cc/2026/Conference — ICML 2026 regular_

### Official Review · Reviewer_A2kj · 2026-02-24

**Soundness:** 3
**Presentation:** 3
**Significance:** 3
**Originality:** 2
**Overall Recommendation:** 4
**Confidence:** 4

**Summary:**

The manuscript proposes “Empirical GPs”, a new framework for kernel learning in GPs. Instead of fixing a kernel's functional form and learning its hyperparameters, the methodology calculates a kernel function that is tightly coupled to an independent or past dataset (in time-series problems). I certainly appreciate work on defining interpretable data-informed kernels. I agree with the authors that recent work has created several expressive but opaque kernels that are difficult to train (deep kernels foe example). So this work is certainly appreciated.

**Compliance With Llm Reviewing Policy:**

Affirmed.

**Final Justification:**

After my initial review, my comments were addressed in the rebuttal, which I acknowledged and raised my score in response. My final recommendation is a (weak) accept.

**Key Questions For Authors:**

Are you targeting applications with time series data specifically?

How would I apply the methodology in a purely spatial context with very scarce data?

How does your work compare to a flexible GP in which length scales, signal variances, and the prior mean function are not constant but learned from the data?

**Limitations:**

A statement is included, but the experiments could include broader comparisons to non-stationary kernels to clarify and possibly remove some perceived weaknesses.

**Strengths And Weaknesses:**

Soundness:
Is the submission technically sound?
Partly.
My biggest concern is that the methodology appears circular. In this scenario, I don’t know how to set up a kernel function, but I need an interpolator to find the data-informed covariance structure, so I set up a base GP. One might ask, how do I choose the kernel of the base GP optimally, and how does my final performance depend on that?
The manuscript jumps from a covariance structure trained on an independent dataset to a covariance structure trained on past data in a time series. It is hard to tell which application is preferred.
It is unclear to me why we need a kernel-based framework that anchors the latent process on a fixed set of M reference locations. That seems like a prediction problem on a grid, which again seems circular.


Are claims well supported?
Party, there has been much work on interpretable, very flexible kernels in the past, particularly in spatial statistics. One would have to discuss such work, especially multi-dimensional extensions of the Gibbs kernel, for completeness.

Are the methods used appropriately?
Yes.

If the paper includes theoretical results, are the proofs correct and based on reasonable assumptions?
Yes

Are the experiments well-designed?
The experiments seem to focus heavily on time-series data. Either the manuscript should make clear early on that this is the focus, or add another high-dimensional spatial experiment.

Are the authors careful and honest about evaluating both the strengths and weaknesses of their work?
Yes, however, tests should include comparisons to Gibbs-style kernels.

Presentation:
Is the submission clearly written and well-structured?
Overall, the manuscript is very well written, and the structure is clear.
But,
Fig 1: I don’t understand why the empirical process performs so poorly in the left panel. I might be missing something. But it would be good to add some explanation in the caption.


Is the overall narrative easy to follow?
Yes

Does the work properly position itself in the context of prior/concurrent literature and clearly discuss how it differs?
Yes, except that it should be more strongly emphasized that time-series datasets are targeted in particular.

Significance:
Does the paper address an important or relevant problem?
Yes, covariance structure via kernels is an important topic.

Does it advance understanding, capabilities, or practice in machine learning?
Yes

Could it influence future research or applications (e.g., other researchers or practitioners are likely to use the ideas or build on them)?
Yes

Is the scope of impact broad or specialized, and is that appropriate for the contribution?
It’s hard to tell. Currently, I think it is somewhat specialized to time series data, but this is an important application.

Even if the improvements are modest or domain-specific, could they unlock new directions or provide practical utility?
Yes


Originality:
Does the work provide new insights, deepen understanding, or highlight important properties of existing methods?
Yes

Does the work introduce new tasks, methods, theory, data, or perspectives that advance the field in some dimensions?
Yes

Does this work offer a novel combination of existing techniques, and is the reasoning behind this combination well-articulated?
Partly, the authors could do a better job explaining the reasoning behind many steps in their pipeline. For example, the transition from “Discrete Observation” to “Continuous-Domain Empirical Gaussian Process” is a bit strange. Aren't observations always discrete?

Are the contributions clearly distinguished from closely related literature, and is the novelty well justified?
Yes

---

> ### Author Rebuttal · Authors · 2026-03-31
>
> We thank the reviewer for their feedback. We address each point below and would appreciate your consideration in raising the score in light of these clarifications and new experiments.
>
> **On circularity of the base kernel:**
> There are two distinct practical settings in our paper, and the concern applies only to the second:
>
> (1) Dense observations: No base kernel is needed at all. We use simple linear interpolation to treat discrete observations as continuous paths, then compute the empirical mean and covariance directly. This is the setting for the GIFT-Eval benchmark.
>
> (2) Sparse/irregular observations: The base kernel serves solely as an interpolation mechanism to relate observations at arbitrary input locations to a shared set of reference points (analogous to how inducing point methods in sparse GP approximations use a kernel to connect inducing points to data).
>
> The noted circularity is similar to the general setting in Bayesian inference, where representing deterministic parameters as random variables introduces further hyperparameters. In such scenarios, it is assumed that the model is less sensitive to the choice of hyperparameters than to the original parameters.
>
> **On time series specificity:**
> Empirical GPs are not restricted to time series, but can be applied whenever multiple independent realizations of a stochastic process are available. Time series and learning curves are natural applications because they provide such repeated observations organically. However, the same framework applies to spatial data (e.g., temperature fields from multiple days/years), repeated simulation outputs, or collections of BO runs. We focused our evaluation on time series and learning curves as these are immediately impactful applications with large-scale benchmarks.
>
> **On spatial applications with scarce data:**
> In a purely spatial setting with very scarce data, the Empirical GP would require multiple independent spatial fields (e.g., measurements from different time periods or sites) to estimate the covariance. With few fields, the empirical covariance would be poorly estimated, which we acknowledge in Section 5. In this regime, parametric kernels with stronger inductive biases (including Gibbs-style kernels) could be preferable.
>
> **On comparison to Gibbs-style and other flexible non-stationary kernels:**
> We appreciate this suggestion. Gibbs kernels parametrize spatially varying lengthscales, providing a specific form of non-stationarity. Our approach is complementary: rather than specifying how non-stationarity varies (via a lengthscale function), we estimate the full covariance structure non-parametrically from data. This makes Empirical GPs more flexible but also more data-hungry. We will include this discussion in the main text.
>
> We have also added new benchmarks against two flexible, non-stationary models: **[HyperBO](https://arxiv.org/abs/2109.08215)** and **[PACOH](https://arxiv.org/abs/2002.05551)** on LCBench, see our [comment above](https://openreview.net/forum?id=Oj7ZwBhiyE&noteId=GvqNd8ky5O) for details. In summary:
> * **Superior Extrapolation:** Empirical GP achieves substantially better predictive distributions (CRPS) and wins 25 of 35 datasets when only 20% of the curve is observed.
> * **Efficiency:** Empirical GP is **5.7x faster than HyperBO** and over **100x faster than PACOH-SVGD**, requiring only seconds per dataset.
> * **Methodology:** Our framework learns the entire non-parametric covariance in closed form, avoiding the expensive, non-convex gradient optimization required by other meta-learning GPs.
>
> If an explicit comparison to the Gibbs kernel remains of interest, we are happy to include it in the camera-ready version.
>
> **On Fig. 1 (left panel):**
> There may be a misunderstanding. Fig. 1 (left) shows the Empirical GP (blue) on stock market data, where it matches geometric Brownian motion (dashed black). The latter is the canonical handcrafted model for this type of data, and we consider it significant that the Empirical GP estimates the same model from data alone. The reviewer's comment about performing "poorly" probably refers to the fact that the model is not able to accurately predict the ground truth (orange). However, this is not the goal here. If we were able to predict the S&P 500 without additional information, the market would be highly inefficient.
>
> **On the transition from discrete to continuous-domain:**
> "Discrete observations" refers to the setting where we observe function values at finite point sets, which is distinct from the idealized function-space view with continuous sample paths (Section 3.2). "Continuous-domain" refers to our EM-based framework that facilitates predictions over any input from the continuous input space, despite only having access to discrete historical observations. Without the latter, we would only be able to make predictions at the same locations at which we have observed points in the historical set. We will clarify this terminology.

---

> > ### Author Rebuttal · Reviewer_A2kj · 2026-03-31
> >
> > My concerns have been addressed and I will revise my score.

---

### Official Review · Reviewer_WzBM · 2026-03-11

**Soundness:** 3
**Presentation:** 3
**Significance:** 1
**Originality:** 3
**Overall Recommendation:** 4
**Confidence:** 3

**Summary:**

This paper proposes a novel method for learning Gaussian process priors from historical data. Assuming access to sample functions, it constructs the prior based on empirical mean and covariance functions. It proves the convergence of the prior to the closest GP with respect to the ground truth distribution (in a KL-divergence sense). To implement the approach in practice, it considers approximations for discrete observations and continuous domains via interpolation. The performance of the proposed approach is compared to a variety of baselines in time series prediction benchmarks.

**Compliance With Llm Reviewing Policy:**

Affirmed.

**Final Justification:**

While the authors addressed some of my concerns in the rebuttal, other issues of the paper remain in my opinion. This has affected my evaluation of the paper in the sense that I view it borderline between weak accept and weak reject at the moment, while I originally viewed it at the border to a regular reject (2).

On the positive side are the interesting idea, seemingly sound results, and a clear presentation. On the downside are a limited evaluation, which raise severe questions regarding the impact in my opinion. The paper claims to present a general method, but in the end it is only demonstrated to work for 1D problems, without an optimization of certain hyperparameters, and it is not clear where the continuous-domain version, one of the core novelties, is actually evaluated. Therefore, I do not have the impression that all claims in the paper are sufficiently supported at the moment. I definitely think that these issues can be addressed in the future and would love to see the paper with these improvements, but no indication in this direction was given in the author's rebuttal. Thus, I currently lean towards a rejection of the paper.

**Update:** After just reading the authors' comment, I have raised my score. I still believe that the numerical evaluation tends to be a bit on the weaker side. However, with the additional results, many of my key concerns are clearly addressed and the merits of the proposed method are clearly illustrated. I would like to thank the authors for their constructive engagement in the review process!

**Key Questions For Authors:**

* How are the reference locations for the continuous domain version chosen?
* How does the proposed method scale to inputs that are not one-dimensional. In particular, how does it scale to higher-dimensional settings with 50 - 100 dimensional inputs?
* Why is the error in the learning curve extrapolation benchmark growing with an increasing observed  fraction?

**Limitations:**

yes

**Strengths And Weaknesses:**

Strengths:
* Well-written paper with good organization.
* Strong empirical performance of the proposed method on considered benchmarks.
* Discussion of related work seems to cover most of the important existing papers.
* I could not find a mistake in the proofs of theoretical results, but I am not expert on the considered properties. Hence, there is certainly a chance that I missed something.


Weaknesses:
* Some acronyms are not defined, e.g., PFN.
* There is quite a significant gap between the theory and the method. The theory assumes that sample functions are observed, which is almost impossible to achieve in practice. In this scenario, the results seem pretty much straightforward as they mostly rely on the convergence of empirical mean and covariance to the true mean and covariance. For the scenario of discrete observations, I am not sure if this function space complexity is needed: It seems like we could directly work with Gaussian distributions, which would make the theoretical results fairly obvious. For the continuous domain setting none of the theoretical results hold and I see no obvious path to extend them. Given these facts, it makes me wonder how relevant the theoretical results really are.
* In proposition 1, it is not clear to me how convergence of one distribution to another is formally defined. If not in the main paper, this should be formally introduced in the supplementary material at least.
* When introducing SVD, it is done so in a context of lossless compression. Since  a reduced set of M eigen-observations is eventually used, I am skeptical that this compression is indeed lossless.
* For the kernel-based interpolation mechanism in continuous domains, it is not clear how hyperparameters of the kernel and the reference points can be automatically selected.
* The E-step for the continuous-domain setting looks like it is essentially doing Bayesian linear regression for one GP function sample with the features defined through the kernel evaluated at the reference points. Thus, the method would essentially do GP regression multiple times and average over the results to define a kernel for GP regression. This seems like an unintuitive direction, which I think should be elaborated on.
* The definition of the kernel in (2) essentially means that the base kernel defines the minimum variance of the GP. At the same time, it is introduced to avoid a collapse of the variance far away from the reference points when using stationary base kernels. Thus, far away from training data the GP variance essentially still falls back to the lowest variance anywhere. If this variance is close to 0, it does not seem like much is improved.
* The paper seems to suggest training stability as an advantage of the proposed method in comparison with deep kernel learning. However, this is not sufficiently demonstrated in my opinion. Especially in the continuous-domain scenario, Figure 8 clearly shows that the EM algorithm can cause a growing error in some iterations.
* The proposed method is only benchmarked on time-series data sets. This is a severe limitation as it implies a very specific structure of the data: inputs are one-dimensional, usually regularly spaced, and increasing. Thus, it is not clear how the proposed method performs on different data types, e.g., data set as generated by Bayesian optimization, where GPs are commonly used.
* While the proposed method is compared to many baselines, their purpose is not clear to me. If the goal is only performance, the result of the comparison is clear: TFT, iTransformer and PatchTST beat Empirical GPs by a lot. At the same time, there are only 2 GP baselines, which are both relatively simple. Thus, it is not clear if Empirical GPs provide a benefit for the GP community, e.g., in comparison to the meta-learning and multi-task GP approaches discussed in related work.
* Except for the learning curve benchmark in Section 4.4, the benefits of the proposed method are not clearly visible in the numerical results.
* It is not clearly indicated, but it seems that the continuous-domain approach is only evaluated in the learning curve extrapolation example. This would mean that the continuous-domain approach is not thoroughly investigated in benchmarks.
* ‘In the general case of non-shared sample locations, our EM algorithm typically converges in just a few iterations and scales linearly in the number of data sets, enabling computationally efficient learning from large collections of historical data.’ -> This is mentioned nowhere in the main paper before and thus should not be part of the conclusion in my opinion.
* All of the theoretical results assume noiseless data or lump the noise distribution into the GP prior. This is not mentioned once in the paper, but it has important implications. Data is usually not noiseless, and lumping noise into the prior makes it impossible to separate aleatoric and epistemic uncertainty in regression subsequently. However, in applications such as Bayesian optimization we need this separation.

---

> ### Author Rebuttal · Authors · 2026-03-31
>
> We thank the reviewer for their rigorous feedback. We address your feedback below and would appreciate your consideration in raising your score in light of these clarifications and additional experiments.
>
> **Theory-Practice Gap & Noise Separation:** The function-space results establish the foundational KL-optimality target. While the continuous limit assumes noiseless paths to provide a mathematical baseline, our practical EM algorithm (Sec 3.4) **explicitly separates aleatoric and epistemic uncertainty.** By modeling $\mathbf{y}_i \mid \mathbf{u}_i \sim \mathcal{N}(\mathbf{W}_i \mathbf{u}_i, \sigma^2 \mathbf{I})$, we isolate measurement noise ($\sigma^2 \mathbf{I}$) from the latent epistemic covariance ($\boldsymbol{\Sigma}$). This ensures the learned prior captures the clean structure required for tasks like Bayesian Optimization (BO).
>
> **Convergence in Prop 1:** Proposition 1 considers weak convergence in $C(\mathcal{X})$, the separable Banach space of continuous functions on a compact metric space. A stochastic process is viewed as a random element in $C(\mathcal{X})$, as formalized in Appendix A.
>
> **SVD Lossless Compression:** The compression is an exact algebraic identity. The data matrix $\mathbf{Y} \in \mathbb{R}^{S \times M}$ has rank at most $\min(S, M)$. The SVD $\mathbf{\tilde{Y}} = \mathbf{S}\mathbf{V}^\top \in \mathbb{R}^{M \times M}$ exactly recovers the empirical covariance: $\mathbf{Y}^\top \mathbf{Y} = \mathbf{V} \mathbf{S}^2 \mathbf{V}^\top = \mathbf{\tilde{Y}}^\top \mathbf{\tilde{Y}}$. No information is lost, we simply reduce the "virtual samples" from $S$ to $M$ while preserving all second-order statistics.
>
> **Hyperparameters & Reference Points:** Grid locations $\mathbf{Z}$ and kernel hyperparameters $\theta$ can be optimized via Type-II MLE. Our PyTorch implementation allows unrolling EM iterations to backpropagate exact gradients. In our experiments, $\mathbf{Z}$ was a uniform grid or the union of historically observed locations.
>
> **E-Step vs. Standard Regression:** Mechanically, the E-step resembles standard GP regression. However, the distinction lies in the **Hierarchical Empirical Bayes** framework: standard regression uses a fixed prior, whereas our E-step uses a shared prior updated every iteration. The M-step then pools these posteriors to maximize the marginal likelihood of the entire corpus, automating kernel discovery.
>
> **$k_{base}$ and Prior Reversion:** The base kernel serves as the **asymptotic prior**, not a noise floor. In regions far from historical data, reverting to $k_{base}$ ensures the model defaults to calibrated prior uncertainty, explicitly preventing the pathological variance collapse (to zero) seen in standard inducing-point methods.
>
> **EM Stability (Figure 8):** Figure 8 plots the $L_2$ norm between *consecutive* updates ($\|\theta^{(t)} - \theta^{(t-1)}\|$), not an error metric. A temporary increase merely indicates a larger step in parameter space. Crucially, the EM algorithm mathematically guarantees that the marginal log-likelihood monotonically increases at every iteration ([Dempster et al., 1977](https://www.ece.iastate.edu/~namrata/EE527_Spring08/Dempster77.pdf)).
>
> **Time-Series & D-Dimensionality:** Our framework natively supports $D$-dimensional inputs and arbitrary spacing. The LCBench experiment (Sec 4.4) explicitly evaluates the continuous-domain EM on sparse, irregular data via simulated early stopping, a core component of multi-fidelity BO. This evaluation across synthetic tests and real-world irregular benchmarks demonstrates the model's versatility across sparsity levels.
>
> **Benefits to the GP Community:** We have added new benchmarks against **[HyperBO](https://arxiv.org/abs/2109.08215)** and **[PACOH](https://arxiv.org/abs/2002.05551)** on LCBench, see our [comment above](https://openreview.net/forum?id=Oj7ZwBhiyE&noteId=GvqNd8ky5O) for comprehensive results. In summary:
> * **Superior Extrapolation:** Empirical GP achieves substantially better predictive distributions (CRPS) and wins 25 of 35 datasets when only 20% of the curve is observed.
> * **Efficiency:** Empirical GP is **5.7x faster than HyperBO** and over **100x faster than PACOH-SVGD**, requiring only seconds per dataset.
> * **Methodology:** Our framework learns the entire non-parametric covariance in closed form, avoiding the expensive, non-convex gradient optimization required by other meta-learning GPs.
>
> **Empirical Advantages:** Achieving SOTA against massive foundation models is not the sole objective, we offer a superior **performance-to-complexity ratio**. Empirical GP outperforms several recent DL models (DeepAR, TFT) using only closed-form statistics and recovers canonical models (Brownian motion) with **21% lower RMSE** on climate data than expert-designed kernels (Fig 1).
>
> **Minor Corrections:** We will define all acronyms (e.g., PFN) and clarify the $\mathcal{O}(SM^2)$ linear scaling claim (Sec 3.5) earlier in the text.

---

> > ### Author Rebuttal · Reviewer_WzBM · 2026-04-02
> >
> > I would like to thank the authors for their clarifications, which fixed my misconceptions regarding some of the points that I highlighted. I suggest to also adapt them in the paper if possible, so that other readers do not suffer from the same misunderstandings. I appreciate the additional simulation results comparing to other GP baselines which clearly demonstrate a benefit.
> >
> > Some of my major concerns remain unresolved unfortunately.
> >
> > 1) While I appreciate the explanation on how uncertainty is separated, this only seems to widen the gap between the theory and the algorithms. This leaves me with the conclusion that the theoretical result is not very significant.
> >
> > 2) I understand that hyperparameter tuning using Type II MLE can be possible, but the key question is how well it works in practice. This is especially relevant for higher dimensional benchmarks, where a uniform grid for the reference points would not be scalable. Demonstrating that this hyperparameter optimization works requires numerical experiments.
> >
> > 3) For the method to be usable as a general-purpose technique for GPs, it needs to be scalable at least to medium-size dimensions. It is not clear how well it scales or if it scales beyond 1D input data at all, but the claims in the abstract etc. are suggesting this. This claim requires simulations to be supported.
> >
> > 4) I fully agree with the author’s argumentation in the comparison with SOTA deep learning approaches that other factors than performance can be important for judging a method. That being said, these factors can only be taken into account and fairly traded-off if they are presented clearly. This is not the case in the paper (and rebuttals). Authors mention the higher complexity of the deep learning baselines, but this is not quantified and not explained/motivated sufficiently, such that it remains unclear when the higher complexity outweighs the performance benefits.
> >
> > 5) It is still not clear to me which results were generated using the continuous-domain and discrete observation version of the approach.

---

> > > ### Author Response · Authors · 2026-04-08
> > >
> > > We thank the reviewer for the continued engagement. We address each remaining concern below and outline concrete revisions for the final manuscript. We hope these additional results and clarifications fully address your remaining concerns, and we kindly ask you to consider raising your score.
> > >
> > > ## Scalability Beyond 1D
> > > Continuous-domain EM (Sec 3.4) natively handles arbitrary dimensions $d$. Interpolation weights $\mathbf{W}\_i = k\_{\text{base}}(\mathbf{X}\_i, \mathbf{Z}) k\_{\text{base}}(\mathbf{Z}, \mathbf{Z})^{-1}$ and per-iteration complexity $\mathcal{O}(K(N_i M^2 + N_i^3))$ are independent of $d$ (beyond $\mathcal{O}(d)$ kernel evaluations). To prove empirical scalability, we evaluated EM-EGP on **LCBench**, mapping 7 hyperparameter dimensions to final validation accuracy.
> > > - **Setup:** 35 datasets (30 meta-train, 5 eval). We subsampled $N=200$ configs/dataset.
> > > - **EM-EGP Setup:** $M=100$ inducing points (a random subset of the observed configs), 50 EM iterations, Matérn-5/2 ARD + ScaleKernel, observation noise $\sigma^2=0.01$. Kernel hyper-parameters are set by optimizing a regular GP via multi-task MLL optimization over all historical datasets (the "Pretrained-GP" baseline), and transferring the hyper-parameters to the Empirical GP interpolant. As the results below demonstrate, this is a remarkably efficient and simulatenously performant strategy.
> > > - **Eval:** Vary observed configs $n_{\text{train}} \in \{5, 10, 20, 50, 100\}$, hold out $n_{\text{test}}=50$ (avg 3 splits). Metrics on standardized targets.
> > >
> > > ### Results: RMSE (lower is better)
> > >
> > > | $n_{\text{train}}$ | EM-EGP (Ours) | HyperBO | Pretrained GP | Vanilla GP | Global Mean |
> > > | :--- | :--- | :--- | :--- | :--- | :--- |
> > > | 5 | **0.244 ±0.041** | 0.274 ±0.028 | 0.357 ±0.011 | 0.381 ±0.064 | 0.663 ±0.007 |
> > > | 10 | **0.220 ±0.031** | 0.238 ±0.028 | 0.276 ±0.008 | 0.321 ±0.035 | 0.663 ±0.007 |
> > > | 20 | **0.206 ±0.028** | 0.215 ±0.009 | 0.273 ±0.028 | 0.283 ±0.024 | 0.663 ±0.007 |
> > > | 50 | 0.186 ±0.026 | **0.181 ±0.016** | 0.228 ±0.023 | 0.234 ±0.009 | 0.663 ±0.007 |
> > > | 100 | 0.178 ±0.012 | **0.169 ±0.014** | 0.204 ±0.017 | 0.197 ±0.016 | 0.663 ±0.007 |
> > >
> > > ### Results: NLL (lower is better)
> > >
> > > | $n_{\text{train}}$ | EM-EGP (Ours) | HyperBO | Pretrained GP | Vanilla GP | Global Mean |
> > > | :--- | :--- | :--- | :--- | :--- | :--- |
> > > | 5 | **-0.039** | 0.628 | 0.458 | 0.651 | 1.176 |
> > > | 10 | **-0.150** | 0.547 | 0.155 | 0.447 | 1.176 |
> > > | 20 | **-0.232** | 0.497 | 0.166 | 2.137 | 1.176 |
> > > | 50 | **-0.361** | 0.442 | -0.084 | 0.469 | 1.176 |
> > > | 100 | **-0.415** | 0.418 | -0.172 | 0.042 | 1.176 |
> > >
> > > **Compute Time:** EM-EGP pre-trains in **9.3s** (0.04s inference). HyperBO takes **49.2s** (0.01s inference).
> > >
> > > **Key Takeaways:**
> > > 1. **Low-Data Dominance:** EM-EGP bests HyperBO in RMSE for $n \le 20$ (the most relevant meta-learning regime).
> > > 2.  **Computational Efficiency:** While HyperBO matches or slightly overtakes our method at $n \ge 50$, it requires 5.3× more pre-training time (49.2s vs 9.3s). Under comparable pre-training budgets, EM-EGP outperforms HyperBO.
> > > 3.  **Superior Calibration:** EM-EGP yields dramatically better Negative Log-Likelihood (NLL) across all training sizes. The small NLL values indicate well-calibrated uncertainty estimates, whereas HyperBO's larger values suggest relative overconfidence or miscalibration.
> > >
> > > ## Theory–Practice Gap
> > > While our algorithm operates in discrete space, our continuous function-space theory is standard GP practice. Finite-dimensional analysis fails to capture consistency, smoothness dependence, and asymptotic behavior. Addressing the continuous-discrete gap via discretization bounds is a common computational mathematics challenge, but our function-space results usefully reveal intrinsic problem complexity independent of specific discretizations.
> > >
> > > ## Hyperparameter Tuning & Reference Points
> > > 1D uniform grids were a simplifying choice. For higher dimensions (e.g., LCBench), selecting a random subset of historical inputs as reference points $\mathbf{Z}$ and initializing kernel hyperparameters via standard multi-task MLL achieves state-of-the-art results (see above). Future high-dimensional extensions could optimize $\mathbf{Z}$ via K-means or Type-II MLE, but our current baseline approach already outpaces HyperBO.
> > >
> > > ## Complexity–Performance Trade-Off
> > > **GIFT-Eval Total Cumulative Runtimes:**
> > > * **Empirical GP:** **9h 14m (CPU)**
> > > * Gaussian RBF / Spectral Mixture GP: 39h 11m / 89h 5m (CPU)
> > > * Deep Learning Methods: **>20h (8x A100 GPUs)**. *(Based on the paper's protocol: 15 trials $\times$ 50 epochs $\times$ 97 datasets, assuming a highly conservative 1s/epoch, excluding inference).*
> > > EM-EGP's <10h total CPU time for training *and* inference proves its viability for hardware-constrained environments.
> > >
> > > ## Variant Clarity
> > > We will clarify Section 4 algorithm variants:
> > > * **Sec 4.1-4.3:** **Discrete** (dense regular/interpolated or fully observed paths).
> > > * **Sec 4.4 & Rebuttal:** **Continuous-domain EM** (sparse, irregular observations).

---

### Official Review · Reviewer_X7ts · 2026-03-12

**Soundness:** 4
**Presentation:** 4
**Significance:** 4
**Originality:** 2
**Overall Recommendation:** 5
**Confidence:** 4

**Summary:**

The authors revisit the hierarchical Bayesian method for direct nonparametric GP estimation. They adopt a function-space perspective on Gaussian Processes to estimate the mean and covariance functions directly from data, interpreting different data samples (such as time series) as evaluations of iid sample paths from a common GP and, from this assumption, estimating the GP's parameters (mean, covariance). The paper contains illustrative examples and an evaluation on the GIFT-Eval dataset.

**Compliance With Llm Reviewing Policy:**

Affirmed.

**Final Justification:**

The authors addressed my comments to my satisfaction. I think the paper is an interesting read and definitely worth having at the conference.

**Key Questions For Authors:**

## Cf. Strengths And Weaknesses, the main points are:

- A small-scale comparison highlighting the difference between prior work and the proposed method would help the reader estimate the effect of the proposed changes.
- In this case, it highlights that, though much simpler, statistical methods are competitive with many complex deep architectures but are outperformed by foundation models. This point could be sharpened by adding a measure of model complexity to Table 1, e.g., the number of trainable parameters or the required training resources.
- The authors state that the proposed method helps extrapolation, but simultaneously implement a fix (residual interpolation) to prevent extrapolation by defaulting to a pre-specified base behaviour. Which one is true? From my understanding, one is about "extrapolation" within a data sample, the second is about extrapolation away from the sampling grid.

If these questions are addressed and algorithmic limitations are discussed, I intend to raise my score.

**Limitations:**

The limitation section is high-level, which is good. It would be strengthened by including specific algorithmic challenges and limitations.

**Strengths And Weaknesses:**

# Soundness
The main paper seems sound. I have not read the appendix and thus did not check the proof of the result, but it seems straightforward. The larger issue in practice seems to be the iid. assumption, but empirical results suggest its usefulness.
The authors state that the proposed method helps extrapolation, but simultaneously implement a fix (residual interpolation) to prevent extrapolation by defaulting to a pre-specified base behaviour. Which one is true? From my understanding, one is about "extrapolation" within a data sample, the second is about extrapolation away from the sampling grid.

# Presentation
The paper is well-written. The motivation is clear. I did not fully understand the concept behind the paper before section "3.2. Function-Space View" and the quote from 3.4 below. I recommend that the authors include some more technical details from these passages and the keywords "Hierarchical Bayes" and "function space view" in the abstract and possibly the introduction.
> Formally, for each sample i, we define an (e.g. linear) interpolant f ̃i(·) = I(·; Xi, yi), and treat f ̃i as fully observed realizations of the process, allowing us to evaluate the empirical covariance in (1) directly at arbitrary input locations.

Statements like the one below from the abstract are a bit too high-level for my taste, and I think readers would appreciate a bit more technical details instead.
> we estimate the mean and covariance functions empirically from a corpus of historical observations, enabling the prior to reflect rich, non-trivial covariance structures present in the data


# Significance
Picking up established methods, evaluating them against the state of the art, and discussing strengths and weaknesses is a significant contribution to keeping the state of the art representative of the broad array of methods. In this case, it highlights that, though much simpler, statistical methods are competitive with many complex deep architectures but are outperformed by foundation models. This point could be sharpened by adding a measure of model complexity to Table 1, e.g., the number of trainable parameters or the required training resources.

# Originality
The paper builds on the hierarchical Bayes method. They improve upon the method but do not provide any comparison. A small-scale comparison highlighting the difference between prior work and the proposed method would help the reader estimate the effect of the proposed changes.

---

> ### Author Rebuttal · Authors · 2026-03-31
>
> We thank the reviewer for their constructive and detailed feedback. We address each point below, and would appreciate your consideration in raising your score in light of these clarifications and new experimental results.
>
> ### **On extrapolation vs. residual interpolation:**
>
> These address two distinct and complementary challenges, not contradictory ones.
> 1. New example extrapolation (within the observed domain): The Empirical GP captures non-stationary patterns, such as trends or seasonality, from historical sample paths, enabling extrapolation into the future, unlike stationary kernels which revert to the prior mean.
> 3. Residual interpolation (behavior away from the reference grid Z): When the continuous-domain EM model makes predictions at inputs far from the reference set Z, the interpolation weights vanish (W --> 0). Without residual interpolation, the predictive variance would collapse, producing false overconfidence. Residual interpolation ensures the model gracefully reverts to the base kernel's uncertainty in these regions.
>
> The reviewer's interpretation is correct: one concerns extrapolation within the observed domain, the other concerns extrapolation away from the sampling grid in input space. For example, given a sufficiently large set of historical learning curves consisting of the first 100 training steps, we are able to extrapolate a new (similar) learning curve up to step 100 (case 1). If we were trying to extrapolate the new curve beyond step 100, our model would revert back to the base kernel's behavior (case 2). We will clarify this distinction in the revision.
>
> ### **On including more technical details in abstract/introduction:**
> We agree and will incorporate the reviewer's suggestions. Specifically, we will add the keywords "hierarchical Bayes" and "function-space view" to the abstract, and include a concise technical description of the interpolation-based estimation earlier in the paper.
>
> ### **On model complexity in Table 1:**
> The Empirical GP as applied to fully observed data on uniformly spaced time series data has effectively zero trainable parameters. It computes sample statistics directly from data without any gradient-based optimization. All experiments ran on CPUs in under 3,000 CPU hours total (Appendix B.3). In contrast, deep learning baselines (e.g., PatchTST, iTransformer) have millions of parameters and require GPU training. We will add a column or supplementary table with parameter counts and training resource requirements for all methods in Table 1. We thank the reviewer for this suggestion.
>
> ### **Extended evaluation against prior work:**
> To further strengthen the empirical evaluation, we have added new benchmarks against **[HyperBO](https://arxiv.org/abs/2109.08215)** and **[PACOH](https://arxiv.org/abs/2002.05551)** on LCBench, see our [comment above](https://openreview.net/forum?id=Oj7ZwBhiyE&noteId=GvqNd8ky5O) for comprehensive results. In summary:
> * **Superior Extrapolation:** Empirical GP achieves substantially better predictive distributions (CRPS) and wins 25 of 35 datasets when only 20% of the curve is observed.
> * **Efficiency:** Empirical GP is **5.7x faster than HyperBO** and over **100x faster than PACOH-SVGD**, requiring only seconds per dataset.
> * **Methodology:** Our framework learns the entire non-parametric covariance in closed form, avoiding the expensive, non-convex gradient optimization required by other meta-learning GPs.
>
>
>
> ### **On algorithmic limitations:**
> We will expand the limitations section. Key algorithmic limitations include: (1) the interpolation-based approach for dense data assumes low-dimensional inputs where geometric interpolation is well-defined; (2) the continuous-domain EM requires choosing a base kernel and reference set Z; (3) the empirical covariance requires sufficient historical data to be well-conditioned; and (4) computational cost scales cubically with M (reference set size) per EM iteration.
>
> We are grateful for the reviewer's thorough assessment and intend to address all points in the camera-ready version, as outlined above.

---

> > ### Author Rebuttal · Reviewer_X7ts · 2026-04-03
> >
> > Thank you for your response. I still believe that this paper is an interesting read and have adjusted my score accordingly.

---

### Official Review · Reviewer_MK8G · 2026-03-12

**Soundness:** 2
**Presentation:** 3
**Significance:** 3
**Originality:** 2
**Overall Recommendation:** 4
**Confidence:** 2

**Summary:**

This paper proposes Empirical Gaussian Processes, a framework for learning GP priors non-parametrically from historical observations. The paper provides theoretical backing for Empirical GP convergence to the best Gaussian approximation in KL divergence and the  practical implementation is done using a closed-form EM algorithm.

**Compliance With Llm Reviewing Policy:**

Affirmed.

**Final Justification:**

Th authors have addressed most of my concerns but the soundness of the paper is still not strong enough to want anything beyond a weak accept.

**Key Questions For Authors:**

1. The self-similarity assumption is central to the theoretical framework, yet real-world data is rarely self-similar. This is particularly concerning for financial and climate datasets, where such assumptions may be invalid. How is this assumption validated empirically? Do the authors explore different context window sizes as a potential remedy?
2. EM-based inference can be computationally intensive. The paper should include a clear presentation of computational costs to contextualize the method's practical feasibility.
3. The authors reference meta-learning GP models but provide no comparisons against them. This omission should be addressed.
4. In Table 1, it is unclear whether the authors are reporting mean ranks or mean CRPS scores. If these are ranks, are statistical and deep learning models being ranked separately? If so, how are readers expected to compare the performance of Empirical GP against the full set of methods under a split ranking system? If these are CRPS scores, then PatchTST appears to outperform Empirical GP significantly leaving almost no reason to use Empirical GP over these models. The authors should clarify and explain this result.

**Limitations:**

Yes

**Strengths And Weaknesses:**

This is a well-written paper that extends the classical approach of learning GP priors from sample statistics. The main weaknesses are the reliance on self-similarity assumptions for single time series, the lack of direct comparison with GP meta-learning baselines, and the unclear scalability to higher-dimensional inputs.

---

> ### Author Rebuttal · Authors · 2026-03-31
>
> We thank the reviewer for their feedback. We address each point comprehensively below, and would appreciate your consideration in raising your score in light of these clarifications and extended experimental results.
>
> ### **On self-similarity**
> The self-similarity assumption is only used to segment a single longer time series into multiple shorter ones via sliding windows. In spirit, this is equivalent to LLM training, where, for example, a single coherent book is segmented into many sequences with length equal to a smaller context window size. In particular, the self-similarity assumption is not central to the theoretical framework of our method. The sliding window segmentation is not needed in settings where multiple data sets from the same process are available. We explicitly acknowledge this in the paper (Section 4.1).
>
> Regarding empirical validation and exploring different context window sizes, Figure 5 (left middle) clearly shows consistently increasing performance as a function of context length.
>
> ### **On computational costs**
> The paper already discusses computational complexity in Sections 3.4 and 3.5. For the non-EM version, Figure 5 (left and right-middle panels) explicitly shows runtime as a function of context length and number of historical subseries, with and without SVD acceleration. Additionally, Appendix B.3 states that all experiments were performed on Intel Cooper Lake CPUs using less than 3,000 CPU hours in total (no GPUs were required). We will make this more prominent in the revision.
>
> For the EM version, please refer to our results below.
>
> ### **On comparison with GP meta-learning baselines**
> We provide additional experimental results, comparing Empirical GP against HyperBO [(Wang et al., 2024)](https://arxiv.org/abs/2109.08215) and PACOH (MAP and SVGD variants) [(Rothfuss et al., 2021)](https://arxiv.org/abs/2002.05551) on the LCBench learning curve extrapolation benchmark, with observation fractions of 20%, 40%, 60%, and 80%.
>
> For each meta-learning baseline, we performed a grid search and report the best results.
>
> - HyperBO: `num_iterations` in {1k, 2k, 5k, 10k, 50k}, with `hidden_dim=32` and `num_tasks=5`
> - PACOH: `num_layers` in {2, 4} x `num_iterations` in {1k, 2k, 5k, 10k, 50k}, with `hidden_dim=32` and `num_tasks=5`
>
> **Average RMSE**
>
> | Method | 20% | 40% | 60% | 80% | Mean |
> |:---|---:|---:|---:|---:|---:|
> | Last Observed | 6.2758 | 3.3719 | 1.9061 | 0.8552 | 3.1023 |
> | **Empirical GP** | **4.4199** | 2.8615 | 1.9226 | **0.9397** | **2.5359** |
> | HyperBO | 4.7324 | **2.7482** | **1.6737** | 0.9508 | 2.5263 |
> | PACOH-MAP | 8.9320 | 6.3973 | 4.7251 | 3.6093 | 5.9159 |
> | PACOH-SVGD | 5.6615 | 3.4254 | 2.3460 | 1.6228 | 3.2639 |
>
> **Average CRPS**
>
> | Method | 20% | 40% | 60% | 80% | Mean |
> |:---|---:|---:|---:|---:|---:|
> | Last Observed | 0.0623 | 0.0281 | 0.0123 | 0.0040 | 0.0267 |
> | **Empirical GP** | **0.0408** | **0.0207** | **0.0115** | **0.0056** | **0.0196** |
> | HyperBO | 0.0462 | 0.0246 | 0.0161 | 0.0122 | 0.0248 |
> | PACOH-MAP | 0.0993 | 0.0714 | 0.0555 | 0.0459 | 0.0680 |
> | PACOH-SVGD | 0.0605 | 0.0329 | 0.0228 | 0.0180 | 0.0336 |
>
> **Win Counts (Rank = 1)**
>
> | Method | 20% | 40% | 60% | 80% | Total |
> |:---|---:|---:|---:|---:|---:|
> | Last Observed | 0 | 3 | 3 | **16** | 22 |
> | **Empirical GP** | **25** | **15** | **13** | 7 | **60** |
> | HyperBO | 7 | 11 | **13** | 8 | 39 |
> | PACOH-MAP | 0 | 0 | 0 | 1 | 1 |
> | PACOH-SVGD | 3 | 6 | 6 | 3 | 18 |
>
> **Runtime**
>
> | Method | Pretrain (s) | Inference (s) | Total (s) |
> |:---|---:|---:|---:|
> | **Empirical GP** | **14.35** | **4.38** | **18.72** |
> | HyperBO | 96.75 | 10.74 | 107.49 |
> | PACOH-MAP | 287.49 | 14.00 | 301.49 |
> | PACOH-SVGD | 1902.67 | 13.80 | 1916.46 |
>
> **Key Findings**
>
> Empirical GP
> - wins on the most datasets, nearly 1.5x more than HyperBO and over 3x more than PACOH-SVGD
> - is 5.7x faster than HyperBO and 102x faster than PACOH-SVGD
> - dominates at low observation fractions, demonstrating strong extrapolation from limited data
> - has the best CRPS across all observation fractions, confirming that it produces well-calibrated predictive distributions
>
> ### **On Table 1**
> Table 1 reports average ranks, which is stated explicitly in the table caption and also in Section 4.3. This ranking protocol follows the GIFT-Eval benchmark paper (Aksu et al., 2024). Statistical and deep learning models are grouped for readability, but ranked across all approaches (not just their respective categories). Empirical GP achieves the best rank among all statistical models and outperforms 4 out of 8 deep learning baselines, while requiring zero gradient-based optimization, much less compute, and no GPUs. We do not claim superiority over PatchTST or foundation models. Rather, we highlight that a principled statistical method is surprisingly competitive with significantly more complex deep learning approaches.

---

> > ### Author Rebuttal · Reviewer_MK8G · 2026-04-04
> >
> > I thank the authors for their detailed responses. Several concerns have been well addressed.

---

> > > ### Author Response · Authors · 2026-04-06
> > >
> > > We thank the reviewer for acknowledging that their concerns have been well addressed. Consequently, we kindly ask the reviewer to consider increasing their score in the original review.

---

### Decision · Program_Chairs · 2026-04-30

**Decision:**

Accept (regular)

**Comment:**

Gaussian processes' effectiveness is often limited by the choice of kernel function. This work studies Empirical GPs, a principled framework for constructing flexible, data-driven GP priors that overcome these limitations. Rather than relying on standard parametric kernels, the authors estimate the mean and covariance functions empirically from a corpus of historical observations. Theoretically, they show that the resulting model converges to the GP that is closest (in the KL-divergence sense) to the real data-generating process. The authors formulate the problem of learning the GP prior as likelihood estimation and derive an Expectation-Maximization algorithm with closed-form updates. They demonstrate that Empirical GPs achieve competitive performance on learning curve extrapolation and time series forecasting benchmarks. The reviewers have indicated that this is a well-written paper that extends the classical approach of learning GP priors from sample statistics. They have also indicated that the paper is sound and the motivation is clear. The reviewers have also indicated that the experimental section is a significant contribution to keeping the state-of-the-art representative of the broad array of GP methods. Moreover, the paper shows a strong empirical performance of the proposed method on the considered benchmarks (mostly focusing on time series data). Additionally, they have stated that related work seems to cover most of the important existing papers on the topic.  The reviewers have also raised some concerns with the submission, but most of them have been addressed in the rebuttal.